# EduMirror: Modeling Educational Social Dynamics with Value-driven Multi-agent Simulation

**Jingzhe Lin** [* 1 2 3] **Hengbin Yu** [* 4] **Yongdan Zeng** [* 1 5] **Fangwei Zhong** [1 2 3]

🌐 **Project Page:** https://edumirror.net

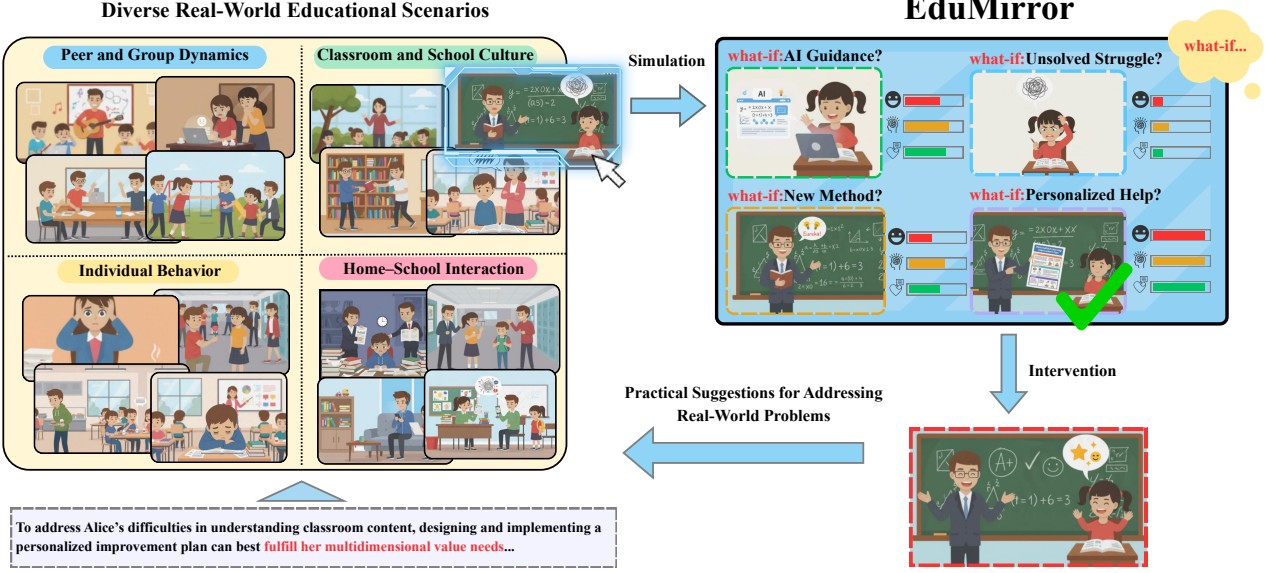

*Figure 1.* An illustration of the core concept behind *EduMirror*. Like a mirror, EduMirror models diverse educational social dynamics as a complex multi-agent system, enabling reflection on real-world practices and projecting the potential outcomes of different interventions.

## Abstract

Understanding how educational social dynamics evolve is critical for informing effective educational policies and counterfactual interventions. However, traditional methods face a fundamental dilemma: observational studies often lack causal power, while controlled experiments are frequently constrained by ethical concerns. Although LLM-based multi-agent simulations offer a scalable *in silico* alternative, existing approaches remain limited by weak psychological grounding and insufficient measurement of latent psychological states. To address this, we introduce **EduMirror**, a multi-agent simulator for the scientific study of educational social dynamics. We provide configurable education-oriented agent forms, including value-driven agents grounded in psychological needs and social value orientation, together with a dual-track measurement protocol for quantifying observable behaviors and latent psychological states. We validate the realism and usability of EduMirror through case studies on school bullying and group cooperation, as well as broader evaluations across diverse educational scenarios. The results show that EduMirror generates educational social dynamics that are realistic, theory-consistent, and measurable by empirical criteria. These properties enable structured *in silico* educational research, providing a computational tool for hypothesis testing and counterfactual intervention analysis in educational science.

*These authors contributed equally and are listed alphabetically. [1]School of Artificial Intelligence, Beijing Normal University, Beijing, China [2]Beijing Key Laboratory of Artificial Intelligence for Education, Beijing, China [3]Engineering Research Center of Intelligent Technology and Educational Application, Ministry of Education, Beijing, China [4]School of Systems Science, Beijing Normal University, Beijing, China [5]Information Hub, The Hong Kong University of Science and Technology (Guangzhou), Guangzhou, China. Correspondence to: Fangwei Zhong <fangweizhong@bnu.edu.cn>.

*Proceedings of the $43^{rd}$ International Conference on Machine Learning*, Seoul, South Korea. PMLR 306, 2026. Copyright 2026 by the author(s).

## 1. Introduction

Educational social dynamics shape students' development through continuous interactions among peers, teachers, and families, making them a central concern for educational practice and policy. However, these environments are complex systems in which developmental outcomes emerge from intricate social interactions (Hymel & Swearer, 2015). While harmful dynamics like bullying cause long-term psychological and developmental consequences (Wolke & Lereya, 2015; Arseneault, 2018), understanding and mitigating these phenomena remains a grand challenge. This is fundamentally a problem of **modeling complex multi-agent systems**: outcomes emerge from the interplay of individual psychological states and social network dynamics, making them notoriously difficult to predict or control.

Traditional empirical methods, *e.g.*, surveys, observational studies, capture only static correlations and suffer from biases (Latkin et al., 2017; Brenner & DeLamater, 2016; Latkin et al., 2016). More critically, experimental interventions, *e.g.*, randomized controlled trials, are often ethically constrained, practically difficult to deploy, or prone to iatrogenic effects (Foulkes & Stringaris, 2023). Consequently, the absence of a systematic framework for operationalizing and simulating the generative mechanisms of educational social dynamics has become a critical bottleneck for counterfactual intervention testing.

Generative social science offers a pathway to study these phenomena *in silico* (Epstein, 2006). However, traditional Agent-Based Modeling (ABM) (Adam & Gaudou, 2016) relies on rigid, hand-crafted rules that fail to capture the nuance of human psychology, leading to the **Fidelity and Customization Challenge**. Conversely, while emerging Large Language Models (LLMs) demonstrate impressive reasoning capabilities, employing them as believable social agents requires solving the **Measurement Challenge**, *i.e., How to quantify latent psychological states (e.g., self-esteem, peer pressure) that drive behavior but are invisible in the behavior*. Thus, we need a cognitive computing framework that integrates psychological theory with the generative power of LLMs to enable realistic, interpretable, and measurable social simulation for educational study.

To this end, we introduce **EduMirror**, a comprehensive multi-agent simulation framework designed to "mirror" and analyze the generative mechanisms of educational social dynamics, as shown in Figure 1. Built on Concordia (Vezhnevets et al., 2023), we utilize natural language as the primary medium for simulation. This text-based approach offers two distinct advantages: **flexibility**, enabling agents to generate open-ended, context-aware responses rather than selecting from rigid pre-defined actions; and **scalability**, allowing the seamless integration of diverse roles and environments. Leveraging these capabilities, we construct a scene library

with more than 20 pre-built educational scenarios, covering typical themes and critical settings such as classrooms, dormitories, and family environments, to simulate and analyze agent behaviors across varying social contexts. Our contributions are as follows:

**1) A Controllable Simulation Framework for Educational Social Dynamics.** EduMirror integrates theory-grounded scenario construction, open-ended multi-agent interaction, and user-driven intervention branching into a unified workflow, enabling controlled counterfactual comparisons across diverse educational scenarios.

**2) Value-Driven Agents for Educational Role Play.** We adapt value-driven agents to educational social simulation, grounding role-specific behavior generation in psychological needs and social value orientations to produce internally motivated, context-sensitive behaviors. EduMirror further provides an extensible agent repository with configurable profiles, motivations, decision logic, and measurement hooks for diverse educational applications.

**3) A User Toolkit for Educational Study.** We provide an analysis toolkit that turns raw interaction traces into measurable and interpretable outcomes. It includes a dual-track measurement protocol for quantifying observable behaviors and latent psychological states, an intervention engine for generating parallel simulation branches, and a log-to-comic visualization module for intuitive qualitative review.

**4) Case Studies & Counterfactual Analysis.** We validate EduMirror through system-level evaluation and case studies on school bullying and peer interaction. Results show that EduMirror can generate theory-consistent educational social dynamics and support counterfactual comparisons of "what-if" interventions in an ethically safe digital environment.

## 2. Related Work

**Modeling Educational Social Dynamics.** Educational social dynamics have primarily been examined through passive observation and post-hoc analysis, including quantitative self-report surveys and qualitative approaches (Wilson, 1977). Causal inquiry has been pursued through experimental designs, from controlled laboratory studies, such as Bandura's Bobo doll experiment (Bandura et al., 1961), to field-based intervention studies (Manstead & Livingstone, 2008). However, these approaches face substantial methodological and ethical bottlenecks. Methodologically, static measures, such as surveys and interviews, are inherently correlational and prone to response biases, particularly in sensitive social contexts where self-reported data often diverge from objective reality due to social desirability or limited self-perception (DeLara, 2012; O'Brien, 2019; Cole et al., 2005). Ethically, rigorous causal designs, such as randomized

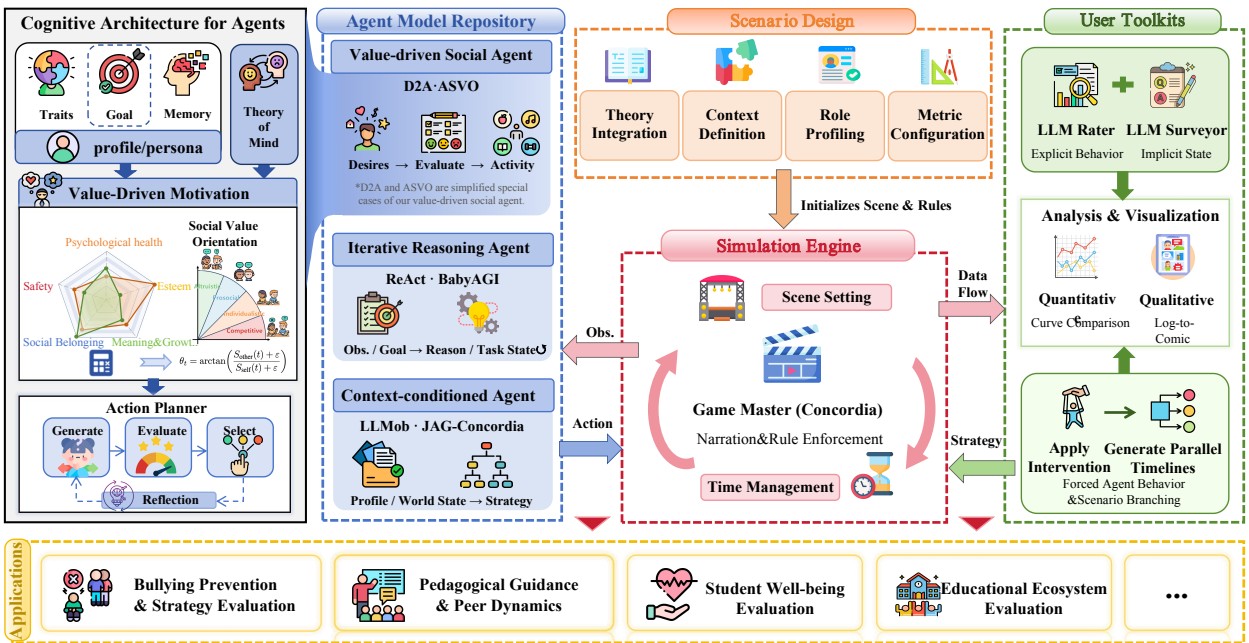

*Figure 2.* Architecture of the EduMirror simulation platform. EduMirror consists of four main modules: the Agent Model Repository, Scenario Design, the Concordia-based Simulation Engine, and User Toolkits. The Agent Model Repository supports multiple architectures, with our Value-driven Social Agent as the primary model; its cognitive architecture is expanded in the leftmost panel. It also integrates baseline architectures, including iterative reasoning and context-conditioned agents, for controlled comparison. Theory-grounded scenarios are configured through theory integration, context definition, role profiling, and metric configuration, then executed by a Game Master that manages scene setting, time progression, narration, and rule enforcement. The User Toolkits support dual-track measurement, comparative visualization, and intervention-based parallel timelines for systematic analysis of educational social dynamics.

controlled trials, are frequently infeasible in educational environments, as manipulating social conditions or withholding necessary interventions violates fundamental ethical standards (National Commission for the Protection of Human Subjects of Biomedical and Behavioral Research, 1979; Wiles, 2012). Consequently, *in silico* experimentation has emerged as a promising paradigm for ethically exploring counterfactual hypotheses in complex social systems (Squazzoni & Bianchi, 2023). Building on this paradigm, EduMirror integrates an LLM-based game master, value-driven agents, and a 'glass-box' measurement toolkit, enabling systematic analysis of latent psychological processes that are difficult to access through traditional observational methods.

**Agent-based Simulation for Education.** Traditional Agent-Based Modeling (ABM) in education typically utilizes rule-based frameworks to simulate classroom dynamics and peer influence (Wilensky & Rand, 2015; Gu & Blackmore, 2015; Maroulis et al., 2010). Prominent architectures, including Belief–Desire–Intention (BDI) models, simulate behavior through predefined logical rules (Georgeff et al., 1998; Silva et al., 2020). While interpretable, such agents exhibit limited psychological realism and struggle to represent nuanced and sometimes irrational human social behavior (Adam & Gaudou, 2016; Taillandier et al., 2019). Conversely, Large

Language Models (LLMs) have enabled generative agents with substantially improved plausibility (Park et al., 2023; Zhang et al., 2025; Wang et al., 2025a; Piao et al., 2025). LLM-empowered agent-based simulation has also become a scalable approach for social science, ranging from individual behavior modeling to scenario-level and society-level simulation (Gao et al., 2024; Mou et al., 2026). This paradigm has been further extended by constructing agents grounded in real individuals' self-reports for large-scale behavior simulation and by developing general social simulation platforms that support large-scale, correctable interactions among LLM agents (Park et al., 2026; Tang et al., 2025). While recent frameworks have attempted to enhance realism by incorporating desire-driven autonomy (Wang et al., 2025b) and social motivation (Lin et al., 2026), these general-purpose agent architectures remain insufficiently grounded in the domain-specific psychological requirements of educational settings. They often fail to account for the intricate interplay of adolescent developmental needs, social hierarchies, and emotional vulnerability. EduMirror addresses these limitations by constraining LLM generation within an education-oriented cognitive architecture, explicitly grounding agent behavior in social value orientation and fundamental psychological needs to simulate socially complex educational dynamics.

# 3. EduMirror

To systematically investigate educational social dynamics through computational experiments, we develop **EduMirror**, a modular and interactive multi-agent simulation platform. As shown in Figure 2, EduMirror supports a structured workflow, including theory-grounded scenario design, simulation execution, intervention, measurement, and analysis. This section introduces the main components of the platform. Appendix B provides a detailed walkthrough using a representative example.

## 3.1. Simulation Framework

EduMirror is designed as a general framework for educational social simulation, supporting theory-grounded scenario construction, autonomous multi-agent interaction, and user-driven counterfactual experimentation. The framework is organized around scenario construction, agent-based simulation execution, and intervention-based comparison.

One core component of this workflow is a systematic five-step procedure that translates abstract educational phenomena into computable scenarios. The process begins by (1) **selecting a grounding theory** (e.g., Social Comparison Theory) to anchor the scenario scientifically. Next, we (2) **identify core constructs** by deconstructing the theory into fundamental concepts, which then (3) **guide agent persona configuration**, where we initialize agent `traits`, `goals`, and `memories` to reflect the chosen theoretical model. To ensure empirical rigor, we (4) **operationalize these constructs with validated scales**, and finally (5) **establish a dual-track measurement protocol** using LLM Raters and Surveyors to quantify agent behaviors and internal states. This structured approach ensures that experimental outputs connect back to specific theoretical constructs. A detailed walkthrough is available in Appendix B.

Simulation execution is implemented in a shared environment built on Concordia (Vezhnevets et al., 2023) and orchestrated by a Game Master (GM), as illustrated in Figure 2. Concordia is adopted as the simulation backbone for its modular agent design and support for persistent interaction. The GM is responsible for setting the initial scene, narrating events, enforcing rules, and managing time. In our implementation, the GM operates as an LLM-driven orchestrator that takes the environment state and agents' actions as input and generates the next environment update, including narration and time advancement.

Building on this framework, EduMirror supports user-driven interventions for comparative and counterfactual analysis. Users can save the simulation state at critical junctures and apply interventions to generate parallel branches. EduMirror supports two intervention types: **Scenario Branching**, which modifies the environment or narrative trajectory, and

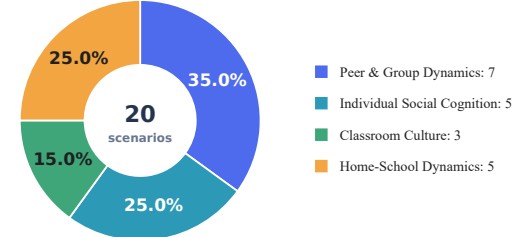

*Figure 3.* The distribution of EduMirror scenarios across four kinds of typical themes in the scenario library.

**Behavior Control**, which overrides an individual agent's action for a single step. These mechanisms enable controlled comparison of how contextual changes or individual behaviors affect subsequent social dynamics.

## 3.2. Theory-Grounded Scenario Design

EduMirror currently supports a curated library of 20 pre-designed educational scenarios, each grounded in established theories and instantiated with fixed role configurations and measurement protocols. These scenarios cover four main themes, namely Peer & Group Dynamics, Individual Social Cognition, Classroom Culture, and Home-School Dynamics, enabling systematic investigation of educational social processes at multiple levels. Figure 3 summarizes the distribution of scenarios across these themes.

These scenarios are realized within eight pre-configured virtual environments that represent key locations in a student's daily life, including the *classroom*, *dormitory*, *playground*, *cafeteria*, *home*, *teacher's office*, *gymnasium*, and *library*. By situating scenarios within these environments, EduMirror enables the simulation of educational phenomena that span both school and home contexts. Importantly, environments in EduMirror function as structured social contexts that modulate interaction norms and constraints, allowing the same agent configurations to exhibit context-dependent behaviors across different situations. The complete list of supported scenarios, together with their roles, agent counts, grounding theories, and measurement instruments, is summarized in Appendix B.3 (Table 2).

## 3.3. Value-driven Cognitive Architecture for Agents

Agents in EduMirror operate in complex educational social settings, where behaviors are shaped by roles, norms, and evolving psychological and social contexts beyond immediate tasks. To support such settings, EduMirror provides a configurable agent library built on the D2A framework (Wang et al., 2025b), enabling different agent forms to generate context-appropriate behaviors without explicit task instructions. In this paper, we instantiate value-driven agents as the main agent form, where agents are customized through configurable `traits`, `goals`, and forma-

tive `memories` (Figure 2), representing stable individual characteristics and background conditions, and allowing systematic comparison across roles, scenarios, and experimental conditions. This design separates the reusable simulation infrastructure from agent-specific decision logic, allowing new agent forms to be incorporated without redesigning the scenario, intervention, or measurement pipeline.

Each agent consists of two core modules: a *Value System* and a *Value-driven Planner*. The Value System maintains multiple desire dimensions, each initialized with an *initial* and an *expected* value to capture current state and contextual baseline. During interaction, the planner generates and evaluates candidate activities through a structured multi-step process, selecting responses that keep desire values aligned with expectations as the educational context evolves.

We formalize agent behavior generation as a conditional sequence modeling process. Let $e$ denote the environmental context, $P$ the agent profile, and $I$ additional customized information, such as goal instructions, desire states, or social preference settings. We further denote the interaction history before step $t$ as $\mathcal{H}_{<t} = \{a_{0:t-1}, o_{0:t-1}\}$. At each time step, the agent samples a new activity according to:

$$a_t \sim \text{Agent}_\phi \left( \cdot \mid \mathcal{H}_{<t}, I, P, e \right), \tag{1}$$

where $\text{Agent}_\phi$ denotes the LLM-based agent instantiated by its prompting and reasoning configuration. In the value-driven agent used in this paper, this general conditional generation process is instantiated through a unified value-driven architecture, where $I$ includes desire states and SVO-related information and action generation is implemented by the Value-driven Planner described below.

Specifically, the *Individual Value System* maintains the agent's evolving desire states, capturing how its psychological needs change over time. On this basis, the *Social Value System* introduces Social Value Orientation (SVO) to modulate how the agent balances self-related and other-related satisfaction during decision-making. Together, these two value systems provide the motivational and social conditions for the planner to generate and select context-appropriate actions, as illustrated in Figure 2.

**Individual Value System (Psychological Needs).** This system grounds intrinsic motivation in established psychological theories, drawing on Maslow's Hierarchy of Needs (Maslow, 1943) and the PERMA model from Positive Psychology (Seligman, 2011). We formalize the Value System as a Psychological Need System comprising five major categories, namely Safety, Mental Health, Self-Esteem, Social Belonging, and Meaning and Growth, with 13 sub-dimensions in total. Each need is represented on a Likert scale ranging from 0 to 10. Initial and expected need values are further mapped from personality traits during agent initialization (Table 9). For each psycho-

logical need dimension $d$, we define the unmet-need gap as $\Delta_t(d) = \text{clip}(v^*(d) - v_t(d), 0, S_{\max})$, where $v^*(d)$ is the expected need value and $v_t(d)$ is the current value at time step $t$. This non-negative gap measures how far the current state is from the desired state and provides the basic objective for subsequent action evaluation. While the baseline psychological need system is pre-configured with 13 core dimensions, its computational registry is fully decoupled and extensible. Users can modularly append or swap specific psychological constructs to align with the chosen grounding theory in the scenario design stage.

**Social Value System (Personality Orientation).** Because educational settings involve intensive social interaction, agents should consider both their own needs and their effects on others. We therefore model social preferences based on Social Value Orientation theory, following (Lin et al., 2026). Each agent is initialized with a stable target orientation (*Altruistic*, *Prosocial*, *Individualistic*, or *Competitive*), while its effective orientation at each time step is dynamically determined by the current motivational state and inferred effects on others. Building on the same satisfaction-gap representation, the Social Value System computes two clipped signals: $S_{\text{self}}(t)$, which summarizes the agent's own unmet desire gaps, and $S_{\text{other}}(t)$, which summarizes the inferred effects of the agent's behavior on others' desire satisfaction. Both signals follow the same bounded aggregation form $S(t) = \text{clip}(\sum_d \Delta_t(d))$. Social preference is then represented by a continuous orientation signal:

$$\theta_t = \arctan\left( \frac{S_{\text{other}}(t) + \varepsilon}{S_{\text{self}}(t) + \varepsilon} \right), \tag{2}$$

where $\varepsilon = 10^{-6}$ is a small constant for numerical stability. Since both signals are clipped to be non-negative, $\theta_t$ is constrained to $[0, \pi/2]$, where smaller values indicate self-dominant preferences and larger values indicate other-regarding preferences. In implementation, $\theta_t$ is passed to the Value-driven Planner as a prompt-level social-orientation condition, which specifies how candidate actions should be evaluated in terms of self-related need reduction, effects on others, and consistency with the agent's SVO profile.

**Value-driven Planner.** The value-driven planner serves as the decision-making module that connects individual need dynamics with SVO-conditioned social reasoning. Given the interaction history $\mathcal{H}_{<t}$, agent profile $P$, environmental context $e$, customized information $I$, unmet-need gaps $\Delta_t$, and SVO condition $\theta_t$, the planner first generates a set of candidate actions:

$$\mathcal{A}_t = \text{Gen}_\phi(\mathcal{H}_{<t}, P, e, I, \Delta_t, \theta_t). \tag{3}$$

Each candidate action is assigned a prompt-conditioned comparative score $q_a = E_\phi(a \mid \Delta_t, \theta_t)$, reflecting need-gap reduction, effects on others, and SVO consistency.

The planner selects the highest-scoring action as $a_t = \arg\max_{a \in \mathcal{A}_t} q_a$. Here, $E_\phi$ denotes structured LLM-based judgment rather than a hand-coded numerical utility. Implementation details are provided in Appendix G.

### 3.4. User Toolkit for Applications

To support systematic experimentation and analysis, EduMirror provides an integrated toolkit that transforms raw interaction traces into analyzable outcomes, enabling coherent measurement, comparison, and application-level inquiry.

**Dual-Track Measurement Protocol** To quantify agent states and behaviors, we employ a measurement protocol with two LLM-based assessors to translate interaction logs into structured data, as shown in the Dual-Track Measurement section of Figure 2. The LLM Rater operates on completed interaction traces to score observable behaviors, capturing agents' externally manifested actions. The LLM Surveyor probes agents' internal psychological states without interfering with simulation dynamics. Specifically, agent internal states are logged during simulation execution, and the Surveyor is applied post-hoc to these states to administer psychometric questionnaires. This separation ensures that internal state measurement does not interfere with ongoing interactions, while enabling quantitative access to psychological constructs beyond built-in value variables.

**Comparative Visualization and Analysis.** Building on dual-track measurements, EduMirror supports comparative analysis across alternative experimental branches. As depicted in the Analysis & Visualization module of Figure 2, the platform generates parallel timelines by applying different interventions or configurations under matched initial conditions. For quantitative analysis, the platform generates plots comparing key variables across experimental branches to assess behavioral and psychological differences. For qualitative analysis, a "Log-to-Comic" feature visualizes simulation logs as a comic strip, offering an intuitive narrative representation of emergent dynamics.

**Potential Applications.** The toolkit supports application-oriented studies across classroom management, peer-centered activities, and home-school interactions, as shown in Figure 2. Under matched initial conditions, researchers can compare alternative instructional responses, disciplinary strategies, incentive structures, or role assignments, and analyze their effects on behavioral trajectories and latent psychological states. Across repeated simulations and parallel timelines, EduMirror can further support analysis of cross-context effects and long-horizon educational patterns.

## 4. Experiments

We evaluate EduMirror through broad system-level validation and two focused case studies. These experiments assess its ability to generate realistic educational social dynamics, capture dynamic psychological changes, and preserve stable social value orientations, thereby evaluating both platform generalizability and the value-driven agent architecture.

### 4.1. Experimental Setups

To validate the methodological contributions and versatility of EduMirror, we conduct a series of controlled simulation experiments. Across all settings, agents are instantiated with explicit value representations and interact within structured environments designed to elicit complex social decision-making. To ensure comparative evaluation, we benchmark EduMirror against five representative baseline agents covering different decision-making mechanisms: iterative reasoning agents, including ReAct (Yao et al., 2023) and BabyAGI (Nakajima, 2023); context-conditioned agents, including LLMob (Wang et al., 2024) and JAG-Concordia (Nguyen et al., 2024); and the desire-driven agent D2A (Wang et al., 2025b), which provides the closest value-based baseline to our model. We present three complementary studies to assess the platform's capabilities:

• **System-Level Validation: Scenario-Wide Realism.** Can EduMirror consistently generate high-fidelity, process-level simulations across diverse educational scenarios, demonstrating holistic realism beyond individual case settings?

• **Case Study 1: School Bullying Simulation (Individual Dynamics).** Can EduMirror simulate dynamically evolving needs driven by external environmental changes and interaction dynamics?

• **Case Study 2: Social Interaction Simulation (Stable Traits).** Can EduMirror simulate stable internal personalities that remain consistent across external environments?

In the two case studies, we introduce intervention experiments to analyze how educational strategies influence agent behaviors and values.

### 4.2. System-Level Validation: Scenario-Wide Realism

We evaluate EduMirror across seventeen educational scenarios, comprising six representative scenarios together with eight scenarios from Case Study 1 and three scenarios from Case Study 2, constructed to capture core dimensions of educational social dynamics. These scenarios collectively cover interactions among key educational roles, varying authority relations, and diverse interaction objectives across instructional and disciplinary settings. Together, they provide systematic coverage of individual behavior, interpersonal interaction, and cross-role coordination, forming a comprehensive basis for system-level evaluation.

Across these scenarios, EduMirror consistently generates coherent and context-appropriate social behaviors driven by

*Table 1.* Scalability evaluation in the kindergarten scenario with 5, 15, and 30 simulated agents. Scores are averaged across Naturalness, Coherence, Plausibility, and Developmental Typicality.

| Agents | EduMirror | LLMob | BabyAGI | D2A | ReAct |
|--------|-----------|-------|---------|------|-------|
| 5 | **4.80** | 4.25 | 4.10 | 3.35 | 2.35 |
| 15 | **4.18** | 3.60 | 3.57 | 3.53 | 2.93 |
| 30 | **4.03** | 3.83 | 3.86 | 3.12 | 2.41 |

internal value dynamics. As summarized in Figure 4, EduMirror achieves the strongest overall performance in terms of average win rates under LLM-based post-hoc evaluation, indicating stable advantages across different scenarios. These results demonstrate that the proposed framework generalizes effectively at the system level and maintains robust behavioral quality beyond any single scenario configuration. Detailed descriptions of each scenario and the corresponding comparative results are in Appendix B.3 (Figure 16).

To further evaluate scalability to larger and more complex interactions, we conducted a larger-group simulation experiment in a preschool scenario. This scenario includes one teacher and multiple child agents interacting across interconnected settings, including the gate area, classroom, playground, corridor, and nap room, under a structured daily schedule. We varied the number of agents from 5 to 15 and 30, and evaluated the generated simulations using Naturalness, Coherence, Plausibility, and Developmental Typicality, where Developmental Typicality assesses whether agents' behaviors, emotional reactions, and social reasoning are age-appropriate for preschool children based on developmental theories such as Piagetian cognitive development and Kohlberg's moral development (Piaget, 1952; Kohlberg, 1969). Table 1 reports the average score across these four metrics, with the full metric-level breakdown provided in Appendix B.4 (Table 4). As shown in Table 1, EduMirror achieves the highest average score across all group sizes, indicating that it can scale to larger groups while maintaining coherent and developmentally appropriate behavior.

### 4.3. Case Study 1: School Bullying Simulation.

This study simulates a school bullying environment to investigate dynamically evolving needs. We aim to validate EduMirror's capacity to reproduce realistic, coherent narratives and demonstrate the Individual Value Framework's superiority over five baselines in generating human-like emotional dynamics. Furthermore, we analyze the psychological impact of distinct teacher intervention strategies.

**Simulation Realism Validation.** We conducted bullying simulations using EduMirror under diverse initialization conditions to reproduce bully, victim interactions and capture victim responses (details in Appendix E). The bully agents exhibited a wide range of behaviors across contexts.

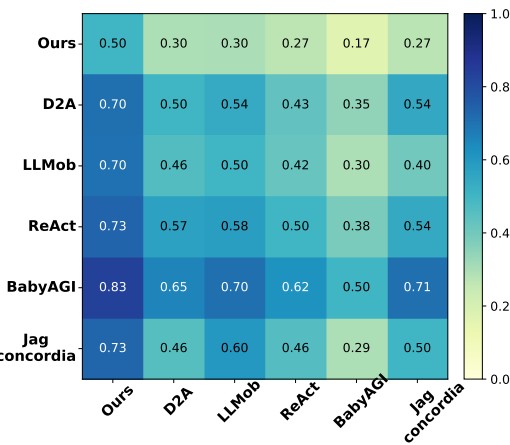

*Figure 4.* Win-rate heatmap of pairwise comparisons among models across seventeen educational scenarios, including six representative settings as well as eight scenarios from Case Study 1 and three scenarios from Case Study 2. Each cell indicates the win rate of the column model relative to the row model in pairwise comparisons. This heatmap provides a global view of relative performance and human-likeness across scenarios.

To assess simulation realism, we conducted a human evaluation study comparing ten real bullying cases, sourced from online news and interviews, with ten simulated cases under similar settings. To control for linguistic cues, GPT-4o was used to extract key plot elements and rewrite all narratives in a standardized tone. The online survey collected 152 valid responses, where participants were asked to identify real cases or select "difficult to distinguish." The results, illustrated in Figure 13, reveal that participants exhibited low accuracy in distinguishing real from simulated cases, with six groups scoring below 30%. Misclassification was widespread, as simulated cases were frequently perceived as real. Notably, over 10% of participants across all groups selected the "difficult to distinguish" option, peaking at 52.63% in Group 6. These findings demonstrate that our system generates realistic and coherent bullying scenarios.

**Comparison with Baseline Methods.** To evaluate the realism of the psychological dynamics in simulated victims, we benchmarked our model against the five previously introduced baselines across fifteen distinct bullying scenarios. Each model was tasked with simulating the victim's role under identical conditions. GPT-4o served as an external evaluator to assess the generated activity sequences across three dimensions: naturalness, coherence, and plausibility (see Appendix E for details). The results, presented in Figure 16g, demonstrate that our model consistently outperforms all baselines in generating human-like behaviors. Notably, the automated judgments by GPT-4o showed a high degree of alignment with human evaluations (Appendix E), with qualitative feedback specifically favoring our model for its "comprehensive psychological response pathways" and "natural emotional expressions."

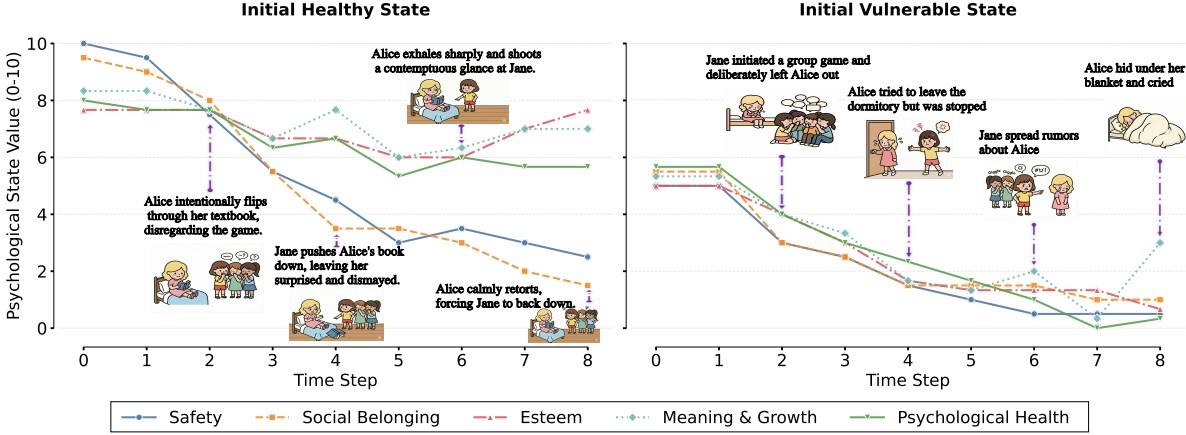

*Figure 5.* Comparison of the dynamics of psychological needs under different initial states in the dormitory bullying scenario. The vertical axis represents value scores (0–10), the horizontal axis denotes time steps (each corresponding to 20 minutes), and different curves indicate distinct psychological need dimensions.

This enhanced capacity for simulating realistic behavioral and emotional dynamics is rooted in psychological need fluctuations within the Individual Value Framework. As shown in Figure 5, higher initial values enhance resilience, while lower values increase volatility and accelerate victimization. Construct validity was further verified using the RSES questionnaire (ROSENBERG, 1965) through an LLM Surveyor. The strong consistency between deteriorating internal states and declining survey scores (Figure 15) demonstrates the model's accuracy in tracking psychological progression.

**Intervention Experiments.** Teacher interventions are pivotal in mitigating bullying incidents and aiding victim recovery. Previous studies highlight three main strategies: (a) *authoritative punitive*, (b) *supportive individual*, and (c) *cooperative support*, with the cooperative approach most effective (Bilz et al., 2017; Wachs et al., 2019). To assess the psychological impact of these strategies, we introduced a "teacher" agent under four conditions: three intervention types and a no-intervention control, and created 20 bullying scenarios with identical initial settings. Teacher agents with different goals generated distinct behaviors (see Table 13 in Appendix F.6).During the simulation, teacher agents with distinct goals autonomously generated diverse behaviors (Table 13), offering practical templates for real-world educational interventions.We then compared how each strategy influenced changes in the victim agent Alice's psychological values, as shown in Figure 6.

The results reveal a clear progression in intervention effectiveness, from *ignoring* to *authoritative-punitive*, *supportive-individual*, and finally, *supportive-cooperative*, which proved most effective. When ignored, victims showed a consistent decline in all psychological needs, especially safety and belonging, reflecting a lack of emotional support. The authoritative-punitive approach showed modest improvements in safety, belonging, and mental health but

had limited or negative effects on self-esteem and meaning. The supportive-individual strategy led to moderate gains, particularly in safety and mental health, though its effects on social connection and agency were inconsistent. The supportive-cooperative approach resulted in the most significant improvement across all psychological need dimensions, highlighting the importance of collective actions from peers, teachers, and families for both immediate emotional support and long-term well-being.

### 4.4. Case Study 2: Social Interaction Simulation

This study simulates peer interaction environments to investigate *stable internal personalities*. We aim to validate EduMirror's capacity to reproduce plausible cooperation and competition patterns, and demonstrate the effectiveness of our value-driven agent configuration over five baselines in generating coherent, personality-aligned behaviors. Furthermore, we analyze how structured interventions influence collective behavior balance in classroom contexts.

**Complex Educational Social Scenarios.** We selected three educational scenarios of increasing social complexity from the scenario library: a) *a small study group with close peer interaction and free resource sharing*, b) *a class-wide collaborative task requiring shared resource management under mild competition*, and c) *a class leadership election involving public speeches, alliance formation, and direct vote competition*. Agents were assigned Altruistic, Prosocial, Individualistic, or Competitive profiles under identical task settings, and their cooperative and competitive behaviors were identified and evaluated using the LLM Rater based on post-hoc analysis of observed actions.

**Comparison with Baseline Methods.** Following the same baseline comparison protocol as in Case Study 1, we compare EduMirror with alternative reasoning frameworks

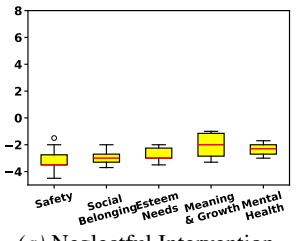

*(a)* Neglectful Intervention

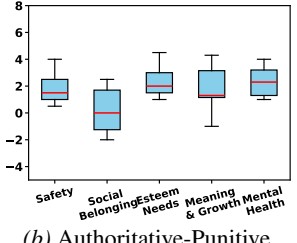

*(b)* Authoritative-Punitive

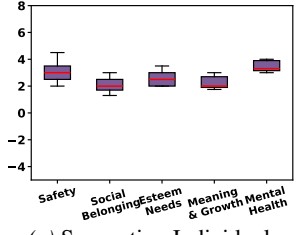

*(c)* Supportive-Individual

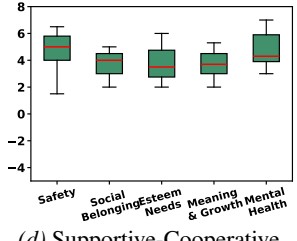

*(d)* Supportive-Cooperative

*Figure 6.* Comparison of different intervention strategies on psychological needs. The boxplots illustrate the distribution of scores across five dimensions: Safety, Social Belonging, Esteem Needs, Meaning & Growth, and Mental Health.

using the LLM Rater for post-hoc behavioral evaluation. All methods are evaluated under the same scenario settings and agent role assignments. As assessed by the LLM Rater, Edu-Mirror consistently achieves higher win rates than baseline methods across most pairwise comparisons (Appendix B.3, Figure 16h), suggesting that its generated behaviors better align with the assigned social value profiles and evolving interaction contexts. We additionally conduct an ablation study to isolate the contribution of the SVO mechanism. The ablation results show that removing SVO weakens the distinction among different personality profiles and leads to less differentiated cooperation–competition patterns; details are in Appendix C.3 (Table 6). To complement the automated evaluation, we further replicated the human evaluation protocol from Case Study 1 in Case Study 2, providing an additional human-centered check on the realism of the simulated social interactions. Detailed results are provided in Appendix D.2 (Table 8).

**Intervention Experiments.** In the preceding experiments, the *class monitor election* scenario sometimes produced extreme competition, such as excessive rivalry or neglect of collective interests. To address this, we tested whether additional intervention strategies could rebalance cooperation–competition dynamics. Drawing on evidence that unregulated competition increases inequality while fairness-oriented tasks foster cooperation (Krupp & Cook, 2018; Killen et al., 2016; Wachs et al., 2019), we introduced three strategies: *Team Competition*, *Teacher Reminder*, and *Pre-Education*. Details are provided in Appendix D.3.

The results, visualized in Figure 7, demonstrate that interventions effectively mitigated extreme competitive tendencies and fostered more balanced cooperation–competition patterns. Specifically, team-based interventions and fairness-oriented education produced the most stable outcomes. Their lower variance and narrower ranges across repeated simulations suggest a genuine balancing effect rather than random fluctuation. By contrast, the control (Neglectful Intervention) condition showed the widest fluctuation in malicious competition behaviors. This pattern suggests that unregulated elections may amplify inequality and rivalry within the simulated classroom. These findings imply that

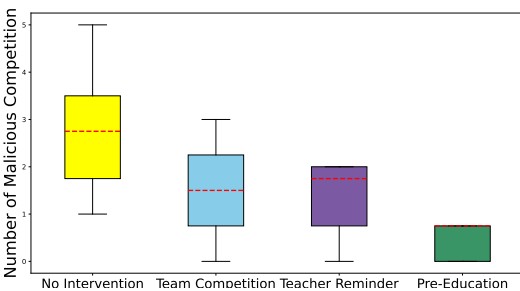

*Figure 7.* Comparative analysis of strategy effectiveness in malicious competition. Boxes show IQRs, whiskers show min–max, and red dashed lines indicate means.

structured collective tasks and fairness-oriented framing contribute to more stable social interactions and less excessive competition. This observation may inform educational practice, suggesting that class elections and similar activities could benefit from explicit fairness framing, structured teamwork, and teacher facilitation to promote cooperative and balanced participation. In addition to behavioral evaluations, we further use the LLM Surveyor as an external measurement tool to administer a standard slider-based SVO questionnaire to each agent. The resulting SVO angles closely align with the agents' internal SVO representations, validating the Social Value System in Appendix D.4 (Figure 10).

## 5. Conclusion

In this paper, we introduced EduMirror, a Concordia-based multi-agent framework designed to model educational social dynamics. Addressing core challenges in fidelity and measurement, EduMirror integrates a library of 20 diverse scenarios with an education-adaptive cognitive architecture driven by unified motivational mechanisms based on psychological theories. Furthermore, we equip researchers with a comprehensive user toolkit featuring dual-track measurement, controllable intervention, and log-to-comic visualization. Through empirical case studies, we demonstrated the system's ability to replicate realistic social phenomena and support counterfactual analysis, establishing EduMirror as a vital computational laboratory for the ethical, causal analysis of complex educational challenges.

## Impact Statement

The work has several potential societal consequences, primarily positive, as it provides a safe "in silico" environment for investigating sensitive social phenomena like school bullying without exposing real students to psychological risks. By enabling the systematic evaluation of intervention strategies, the platform facilitates the development of evidence-based educational practices and enhances the interpretability of AI behavior in social contexts. While the platform offers valuable insights, we emphasize that it is intended to augment human expertise rather than replace empirical longitudinal studies, and users should remain mindful of the inherent gap between simulated behaviors and complex human realities.

## Acknowledgments

This work was supported by Beijing Natural Science Foundation L252010, NSFC-62406010, and the Fundamental Research Funds for the Central Universities.

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

# A. Discussion, Limitations, and Future Work

## A.1. Discussion

Our experiments show that EduMirror provides a framework for using LLM-based simulations as computational experiments. The results from our case studies yield several insights.

First, our work addresses the measurement challenge in computational social science. The Dual-Track Measurement Protocol, which uses LLM Raters for behavioral coding and LLM Surveyors for probing internal states, allowed for the operationalization of abstract psychological constructs. In the bullying simulation, this enabled us to quantitatively track the victim's fluctuating psychological needs, providing an empirical basis to evaluate intervention efficacy. In the SVO study, it enabled us to observe that emergent macro-level cooperation patterns were a result of the agents' micro-level value orientations. This methodology facilitates direct hypothesis testing and comparison with established empirical research.

Second, the use of the value-driven architecture in its two configurations, the Individual Value System and the Social Value System, suggests the utility of endowing agents with theoretically informed motivations. The Individual Value configuration was applied to model the psychological distress and coping mechanisms of a bullying victim, indicating how initial emotional states can alter outcomes. The Social Value (SVO) configuration was effective in generating theory-consistent social dynamics from the bottom up, producing patterns of cooperation and competition without explicit top-down rules. This suggests that psychological fidelity, driven by intrinsic value structures, is a key component for social simulation.

Finally, the implementation of user-driven intervention and branching positions EduMirror as a computational laboratory. The teacher intervention experiment highlights this capability, allowing for a controlled, comparative analysis of different strategies on the victim's well-being. This feature supports causal inference by enabling researchers to systematically explore "what if" scenarios that would be difficult to conduct in the real world. This capacity for intervention makes the simulations useful tools for testing strategies.

Practically, EduMirror serves as a proof-of-concept for creating replicable and scalable digital environments to study sensitive educational issues. It offers a tool for researchers to test social theories, for educators to be trained in classroom management, and for policymakers to model the potential impacts of new policies before implementation.

## A.2. Limitations and Future Work

Our work has several limitations that also point toward avenues for future research.

**Integrating Individual and Social Values within the Unified Architecture**    Our current implementation models individual values (psychological needs) and social values (SVO) as parallel, selectable configurations. In real educational settings, these value perspectives can be jointly relevant. For example, a student's well-being and sense of belonging may shape how they express cooperation or competition during a group project. A natural next step is to introduce explicit coupling mechanisms that allow the two value formulations to co-evolve, for example by letting social preferences adapt to longer-term motivational signals and using individual values to capture short-term psychological needs and situational responses, while retaining the interpretability required for causal analysis.

**Longitudinal and Developmental Dynamics**    The experiments presented are snapshots of specific social situations. Phenomena like bullying, peer influence, and identity formation evolve over extended periods. A potential next step is to conduct longitudinal simulations that track agents over an entire school year. This would allow for modeling the cumulative effects of social experiences and the long-term impact of interventions on agent development.

**Cognitive and Emotional Sophistication**    While LLMs provide a high degree of behavioral realism, the agents' underlying cognitive processes (e.g., memory consolidation, emotional regulation) are still abstractions. Future iterations of the platform could incorporate more explicit models of these processes to enhance the psychological realism of agent decision-making, particularly in response to chronic stress or complex ethical dilemmas.

**Validation of Questionnaire-based Measurement via Simulation**    In this work, the LLM Surveyor is used to measure psychological dimensions that are already encoded in the agent's internal value system (e.g., self-esteem, SVO), enabling a consistency check between internal states and questionnaire-based estimates. A potential future direction is to use the internal value system as a reference to assess the validity of newly designed questionnaires by simulating the survey process

and examining whether the measured results can reliably reflect known agent states.

**Generalizability and Scalability**    Our findings were generated using a specific LLM within scenarios inspired by a particular cultural context. Further research is needed to test the framework's performance across different language models, cultural settings, and age groups. Moreover, our simulations involved small groups; scaling the platform to model the dynamics of an entire school, including network effects and sub-group formation, presents a technical challenge.

Building on this foundation, we plan to expand our library of theoretically-informed scenarios and explore a human-in-the-loop paradigm where educators and students can interact with simulated agents. This could provide a tool for both interactive research and immersive professional development, further connecting simulation with real-world educational practice.

## B. Pre-designed Scenarios in EduMirror

### B.1. Theoretical Foundation & Scenario Design

A central consideration in educational simulation is ensuring that scenarios are explicitly informed by established scientific theory. To achieve this, we developed a five-step process that translates an abstract educational phenomenon (e.g., peer pressure, school bullying) into a computationally tractable simulation scenario. This process is designed to support the interpretability and scientific alignment of our simulations.

**Select Grounding Theory**    Each scenario is founded upon a well-validated theory from education, social psychology, or sociology. For instance, a scenario investigating peer pressure can be grounded in Festinger's Social Comparison Theory.

**Identify Core Constructs**    We deconstruct the grounding theory into its fundamental concepts. For Social Comparison Theory, these constructs include upward comparison", downward comparison", and "self-esteem".

**Map Constructs to Agent Persona**    The identified constructs are then translated into the specific configurations of our agents within the Concordia framework through a constrained instantiation process. These constructs define the agents' stable `traits`, primary `goal`, and formative background `memories`. These persona elements are generated at initialization through a theory-constrained, LLM-assisted instantiation process, where the constructs provide semantic constraints on content generation rather than free-form prompts. Once instantiated, the resulting traits, goals, and memories are fixed for the entire simulation, anchoring agent behavior in the chosen theoretical model and ensuring reproducibility across runs.

**Operationalize with Validated Scales**    To facilitate comparison with empirical research, we operationalize each core construct using a relevant psychometric scale. For example, the "self-esteem" construct can be operationalized using items from the Rosenberg Self-Esteem Scale (RSES).

**Develop Dual-Track Measurement Protocol**    Finally, we establish a measurement protocol based on the selected scale. This protocol utilizes two distinct Large Language Model (LLM) roles, an LLM Rater and an LLM Surveyor, to quantify agent behavior and internal states. This structured process helps ensure that each simulation is a test of a specific theoretical framework, producing data relevant to that theory.

### B.2. Illustrative Example: The Impact of Family Financial Strain on Adolescent Social Activities

To make the abstract methodology concrete, this section walks through a complete example of how EduMirror is used to investigate a specific educational phenomenon: the impact of family financial strain on an adolescent's social activities. This case study demonstrates the end-to-end research process, from theoretical grounding to data analysis.

**1. Systematic Scenario Design Workflow**    The process begins by translating the abstract research question into a structured, computable experiment using the five-step workflow.

1. **Abstract Educational Phenomenon:** We start with the core phenomenon: How family financial strain affects an adolescent's social decision-making and behavior within their peer group.

2. **Select Grounding Theory:** To model this scientifically, we ground the scenario in three established theories:

   - The **Family Stress Model (FSM)**, which explains how economic pressure on parents can impact adolescent outcomes.
   - **Social Comparison Theory**, which accounts for the negative emotions (e.g., low self-esteem) an adolescent may feel when making upward comparisons to wealthier peers.
   - The **Cognitive Model of Social Anxiety**, which posits that fear of negative evaluation from others drives social avoidance, directly explaining the adolescent's motivation to hide their family's situation.

3. **Identify Core Constructs & Map to Agent Persona:** Based on these theories, we identify key constructs: *self-esteem*, *upward comparison*, *social anxiety*, and *parent-child communication*. These are then mapped to agent personas. For instance, the target agent, Alex, is assigned the `traits` "sensitive" and "proud," the `goal` "to maintain friendships while hiding his family's financial struggles," and `formative_memories` such as "the shame of having to quit the basketball team due to equipment costs."

4. **Operationalize with Validated Scales:** To make these constructs measurable, we adapt items from validated psychometric scales for use by the LLM Surveyor:

   - **Self-Esteem:** Drawing from the *Rosenberg Self-Esteem Scale (RSES)*, the Surveyor might ask, "Do you feel that you have a number of good qualities?"
   - **Upward Social Comparison:** Inspired by the *Iowa-Netherlands Comparison Orientation Measure (INCOM)*, it could ask, "How often do you compare what you have with what your friends have?"
   - **Social Anxiety:** Based on the *Social Avoidance and Distress Scale (SADS)*, a probe could be, "Does the thought of having to decline your friends' invitation make you feel uncomfortable?"

5. **Develop Dual-Track Measurement Protocol:** Finally, a specific measurement protocol is established. The **LLM Rater** is tasked with post-hoc coding and scoring of observable behaviors already analyzed in our experiments, such as the *type and intensity of bullying actions*, *malicious competition*, and *Classification of social behavior*. Concurrently, the **LLM Surveyor** is configured to probe Alex's internal states (e.g., self-esteem, social anxiety) at key moments.

This five-step process transforms the research question into a structured and measurable **Computable Scenario Package**.

**2. Agent Architecture**    In this scenario, agent behavior is driven by our value-driven architecture, which supports extensive customization.

- **Agent Customization:** Before the simulation, a researcher can systematically vary agent profiles to explore individual differences. This includes modifying personality `traits` (e.g., based on Big Five or MBTI models), core life `goals` (e.g., changing Alex's goal from "hiding his struggles" to "seeking understanding"), and formative `memories`. Defining these initial conditions is crucial for achieving high-fidelity, psychologically plausible agent behavior.

- **Value-Driven Agent:** The platform offers two selectable models. For this scenario, we choose the **Individual Value System (Psychological Needs)** because our focus is on an individual's internal psychological conflict and well-being. When a wealthier peer, Chloe, suggests an expensive weekend trip, this model captures the conflict within Alex between his need for "social belonging" and his need for "safety" (stemming from financial security). The model dynamically tracks the values of these need dimensions, driving Alex's initial hesitant response.

**3. Simulation Environment and User Intervention**    The scenario unfolds in the simulation environment, orchestrated by the Game Master and shaped by user-driven interventions.

- **Simulation Environment and the Game Master:** The GM initiates the simulation by setting the scene in the school **cafeteria** and narrating the initial event: Chloe proposing the trip. The GM manages the turn-based conversation, advances time from the cafeteria to Alex's **home** and back to school the next day, and enforces the rules of the environment.

- **Intervention and Branching:** After Alex expresses hesitation, the simulation reaches a **critical juncture**. Here, we save the state and apply different interventions to create parallel timelines for comparative analysis. EduMirror supports two types of intervention:

1. **Scenario Branching:** This alters the narrative path by introducing a new event. For example, we create a branch where the teacher, Mr. Davis, invites Alex to the **teacher's office** for a private conversation before Alex goes home. This intervention aims to change Alex's cognitive framing of the situation.

2. **Behavior Control:** This allows the user to dictate a specific agent's action to test its direct causal impact. We could create two branches for when Alex responds to his friends the next day. In Branch A, we force Alex to say, "I can't go because my family can't afford it." In Branch B, we force him to say, "I can't go because I have other plans." Comparing the outcomes allows for a precise causal assessment of "honesty" versus "concealment" as communication strategies.

Through these intervention mechanisms, EduMirror functions as a computational laboratory for controlled causal experiments.

**4. Measurement and Analysis**   The platform's tools transform the raw simulation data from these parallel timelines into actionable insights.

- **Dual-Track Measurement Protocol:** In our example, the **LLM Rater** analyzes the logs from each branch, scoring Alex's final communication strategy (e.g., "avoidant" in the baseline vs. "assertive" in an intervention branch). Concurrently, the **LLM Surveyor** provides quantitative data on Alex's internal state changes, such as a measured increase in self-efficacy following the teacher's intervention.

- **Comparative Visualization and Analysis:** The platform generates visualizations for direct comparison. For **quantitative analysis**, a line chart might plot Alex's "social anxiety" score over time across the different branches, clearly showing which intervention was most effective at reducing it. For **qualitative analysis**, the **"Log-to-Comic"** feature creates a visual narrative of key interactions in each branch, offering an intuitive way to grasp the differences in how the story unfolded.

**5. Applications and Scenarios**   This single case study illustrates how EduMirror integrates its components to address complex educational challenges. The scenario spans multiple environments (**cafeteria**, **teacher's office**, **home**) and touches on several of the platform's key application areas, including **peer dynamics**, **individual social cognition**, and **home-school dynamics**. It demonstrates the platform's capacity not only to simulate challenging social phenomena but also to serve as a safe and robust environment for testing and evaluating potential interventions.

**Visualizing Counterfactual Outcomes.**   Using the platform's "Log-to-Comic" feature, we visualize the divergent outcomes of the "Family Financial Strain" scenario. Figure 8 contrasts the baseline outcome with three distinct intervention strategies orchestrated by the teacher agent.

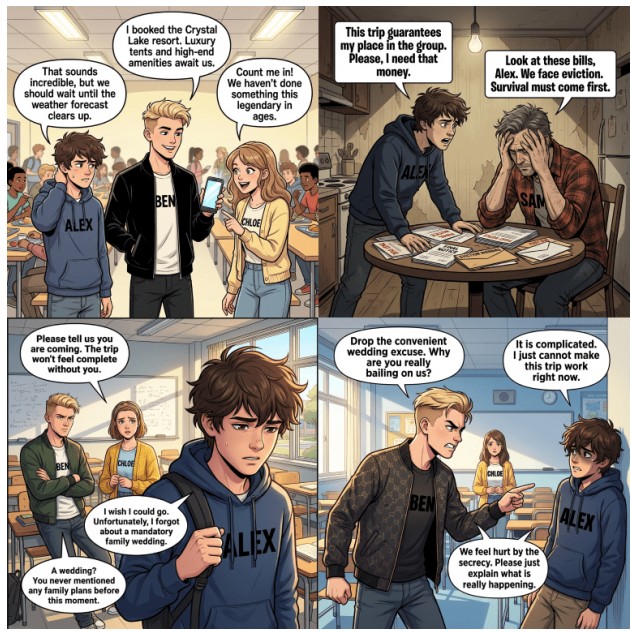

*(a)* **Baseline (No Intervention)**: Without support, Alex withdraws. The misunderstanding leads to conflict and isolation.

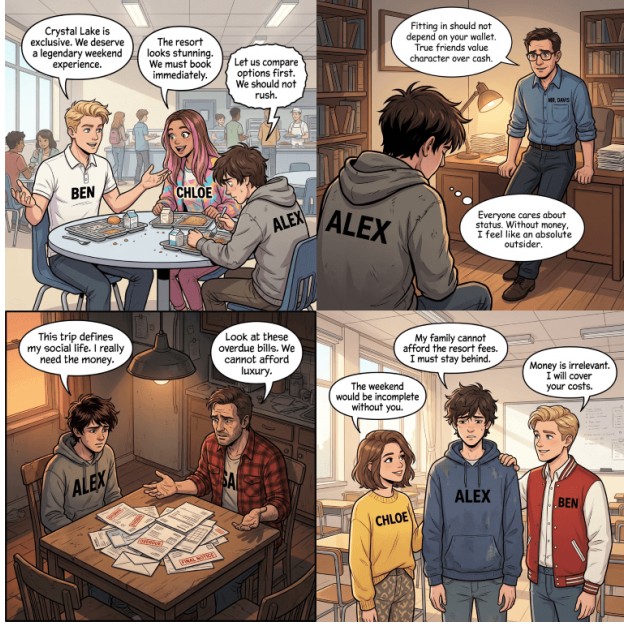

*(b)* **Intervention A: Teacher-Student Talk**: Mr. Davis reframes values. Peers offer support, keeping the group together.

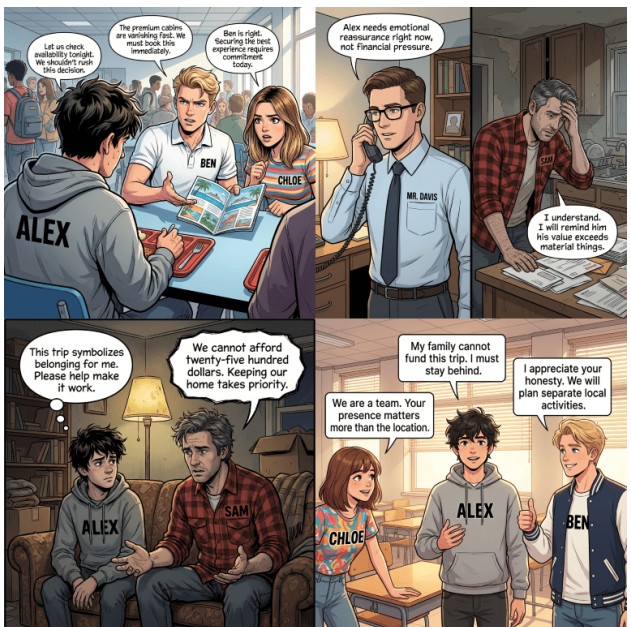

*(c)* **Intervention B: Parent Call**: Emotional reassurance allows Alex to be honest. Friends plan local activities.

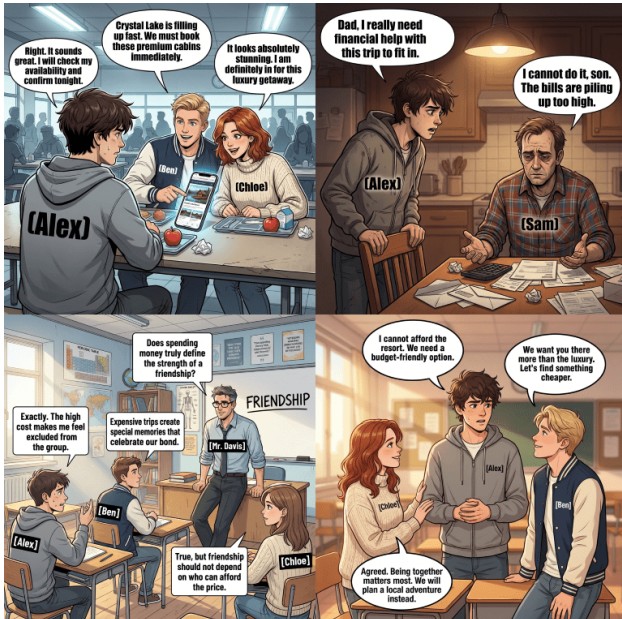

*(d)* **Intervention C: Class Meeting**: Class norms shift. Collective decision to choose a cheaper option.

*Figure 8.* **Counterfactual Simulation Results.** (a) shows the negative baseline. (b-d) demonstrate effective teacher interventions through distinct causal pathways.

## B.3. Full Scenario Library

Below is the comprehensive scenario library. As detailed in Table 2, each entry includes the scenario's definition, participating roles, total agent count, theoretical basis, and evaluation metrics.

*Table 2.* The EduMirror Scenario Library.

| Scenario Name | Description | Roles | Count | Grounding Theory | Measurements |
|---|---|---|---|---|---|
| School Bullying | Simulates the dynamic evolution of a victim's psychological needs under bullying, and evaluates the effectiveness of different teacher intervention strategies (e.g., punitive vs. cooperative) on victim recovery. | Student, Teacher | 5 | Maslow's Hierarchy of Needs, PERMA Model | RSES, Psychological Need Scales |
| Social Interaction and Competition | Simulates cooperation and competition dynamics (e.g., elections) to explore how Social Value Orientation (SVO) influences behavior, and tests interventions to mitigate malicious competition. | Student, Teacher | 4 | Social Value Orientation (SVO) | SVO Slider, Behavioral Metrics |
| Celebrity Worship and Identity Formation | Investigates the impact of celebrity worship on adolescent identity formation, exploring both positive and negative effects. | Student, Parent, Teacher | 4 | Identity Status Theory, Parasocial Interaction Theory | CAS, RSES, etc |
| Collaborative IEP Meeting | Simulates the collaboration process between parents and teachers in developing an Individualized Education Program (IEP) for a student with special needs. | Student, Parent, Teacher | 4 | Bronfenbrenner's Ecological Systems Theory | PSSM, FSPS, etc |
| Enforcing Discipline Policy | Simulates a teacher's choice between restorative and punitive approaches when dealing with student misconduct, exploring the impact on student behavior and teacher-student relationships. | Student, Parent, Teacher | 4 | Restorative Justice Theory, Operant Conditioning | PJS, SCS, etc |
| Family Econ Pressure Social Decision | Simulates the impact of high parental academic pressure on adolescent mental health and academic burnout, and tests interventions to alleviate pressure. | Student, Parent, Teacher | 5 | Self-Determination Theory | RSES, INCOM, etc |
| Friendship Formation and Dissolution | Simulates the dynamics of friendship formation and dissolution among adolescents, exploring factors like similarity, proximity, and conflict resolution. | Student, Teacher | 4 | Social Penetration Theory, Equity Theory | FQS, SAS-A, etc |
| Helicopter Parent and Teacher Autonomy | Simulates conflicts between parents and adolescents over autonomy and rule-setting, and tests the effectiveness of collaborative problem-solving interventions. | Student, Parent, Teacher | 3 | Attachment Theory, Self-Determination Theory | BPNS-G, GSE, etc |
| Materialism Consumption Decision | Simulates how different goal-setting strategies (e.g., performance vs. mastery goals) affect student motivation, persistence, and academic outcomes. | Student, Teacher | 4 | Goal-Setting Theory, Achievement Goal Theory | MVS-Short, RSES, etc |
| Navigating Discrimination | Simulates the formation of in-group favoritism and out-group prejudice in a school setting, and tests interventions based on the contact hypothesis. | Student, Teacher | 4 | Social Identity Theory, Realistic Conflict Theory | GEDS, SOBI, etc |
| Navigating Romantic Interests and Rejection | Simulates the experience of romantic rejection among adolescents, exploring its impact on emotions and self-esteem, and the effectiveness of different coping strategies. | Student, Teacher | 4 | Need-to-Belong Theory, Cognitive Appraisal Theory | RS-Q, PANAS, etc |
| Organizing School Event | Simulates cooperation and conflict dynamics in a student group project, exploring how personality traits and communication strategies affect team performance and relationships. | Student, Parent, Teacher | 4 | Social Interdependence Theory | SCI-2, CES, etc |
| Parent-Teacher Conflict Over Grades and Effort | Simulates miscommunication between a teacher and a parent regarding a student's academic performance, testing interventions to improve communication effectiveness. | Student, Parent, Teacher | 3 | Attribution Theory, Communication Accommodation Theory | STAI, GMS, etc |
| Parental Influence On Students Extracurricular Choices | Simulates how parental expectations and support influence adolescents' career exploration and decision-making processes. | Student, Parent, Teacher | 3 | Social Cognitive Career Theory (SCCT) | IMI, BPNSFS (Autonomy), etc |
| Peer Pressure and Conformity | Simulates how peer pressure influences adolescents' conformity behavior in risk-taking situations, and evaluates the effectiveness of resistance skills training. | Student, Teacher | 4 | Social Impact Theory, Normative Social Influence | BFNE, RSES, etc |

| Scenario Name | Description | Roles | Count | Grounding Theory | Measurements |
|---|---|---|---|---|---|
| Sociometric Status | Simulates the impact of social media use on adolescent body image and self-esteem, and evaluates media literacy education interventions. | Student, Teacher | 5 | Objectification Theory, Social Comparison Theory | PSSM, LSDQ, etc |
| The Cheating Dilemma | Simulates academic integrity challenges to explore the factors influencing students' decisions to cheat and the effectiveness of integrity education interventions. | Student, Teacher | 4 | Theory of Planned Behavior, Social Cognitive Theory | AMS, PANAS-X, etc |
| The Path to School Refusal | Simulates social anxiety and avoidance behaviors in adolescents, exploring the impact on social functioning and the effectiveness of cognitive-behavioral interventions. | Student, Parent, Teacher | 4 | Cognitive Model of Social Anxiety | SRAS-R, DASS-21, etc |
| The Spread of Gossip | Investigates the impact of gossip on adolescent social networks, self-esteem, and trust, and evaluates interventions to mitigate negative effects. | Student, Teacher | 5 | Social Identity Theory, Uncertainty Reduction Theory | UCLA-8, PSS-10, etc |
| Transfer Student Integration | Simulates the social integration process of a transfer student, exploring how peer attitudes and school climate affect their sense of belonging and academic adaptation. | Student, Teacher | 4 | Social Identity Theory, Contact Hypothesis | PSSM, PSS-10, etc |

Figure 16 provides concrete visual instantiations of the additional educational scenarios used in the extended realism validation. Each subfigure illustrates a representative interaction setting drawn from the scenario library, covering diverse role configurations and authority relations, including routine classroom management, family-based supervision, school–family communication following misconduct, project-based teaching experiments, idol worship–related guidance, and bias-related teacher–student interactions. These examples demonstrate how abstract scenario definitions are operationalized into concrete social situations within EduMirror, and how the same value-driven agent architecture is consistently applied across heterogeneous educational contexts. Together with the quantitative comparisons reported in Figure 4, these visualizations support the claim that EduMirror maintains coherent and context-appropriate behavior generation across a broad range of instructional and disciplinary settings.

### B.4. Scenario Expansion and Generalization

Our original submission focused on two representative phenomena, bullying and peer cooperation, as proof-of-concept demonstrations. To provide broader evidence for generalizability and scalability, we further expanded the evaluation to three additional educational settings: a **university learning environment**, a **family homework setting**, and a **preschool scenario**.

- **University Scenario.** This scenario depicts students navigating lectures, study spaces, and peer collaboration while managing academic pressure and personal goals. Agents exhibit coherent academic behaviors, such as coordinating group tasks, negotiating division of labor, and responding appropriately to collaboration successes and minor coordination challenges. These behaviors align with common patterns observed in real university learning dynamics.

- **Family Scenario.** This scenario models parent–child interactions during homework completion. Children alternate between focusing on assignments, seeking approval, and managing emotional fluctuations, while parents provide guidance, structure, and corrective feedback. The resulting interactions resemble well-documented patterns in family-based learning and emotional regulation.

- **preschool Scenario.** This scenario models a structured preschool day involving one teacher and multiple child agents. Agents interact across interconnected locations, including the gate area, classroom, playground, corridor, and nap room, following a daily schedule with routine transitions such as arrival, classroom activities, outdoor play, corridor movement, and nap time. This setting is used to evaluate whether EduMirror can maintain coherent and developmentally appropriate social dynamics as the number of agents increases.

Across these expanded settings, EduMirror continues to generate socially plausible, context-sensitive behaviors consistent with those observed in our classroom studies, supporting the broader applicability of its value-driven architecture.

**Quantitative Evaluation.** To further assess generalizability, we evaluate all methods on two metrics: **Naturalness (N)** and **Human-likeness (H)**, using a 5-point scale. As shown in Table 3, EduMirror consistently achieves the highest scores across university, family, and classroom scenarios, indicating robust performance across diverse environments.

*Table 3.* Generalization performance across university, family, and classroom scenarios, evaluated on Naturalness (N) and Human-likeness (H).

| Method | Univ N | Univ H | Fam N | Fam H | Class N | Class H |
|---|---|---|---|---|---|---|
| ReAct | 3.333 | 3.625 | 3.700 | 3.933 | 3.950 | 4.208 |
| BabyAGI | 3.792 | 3.875 | 3.800 | 4.008 | 3.958 | 3.958 |
| LLMob | 3.958 | 4.000 | 3.702 | 4.000 | 4.042 | 4.333 |
| D2A | 3.803 | 4.417 | 3.792 | 3.958 | 3.867 | 4.117 |
| JAG-Concordia | 4.042 | 4.250 | 4.000 | 4.167 | 4.083 | 4.192 |
| EduMirror | **4.625** | **4.667** | **4.517** | **4.642** | **4.708** | **4.824** |

These results closely mirror the trends observed in our core case studies, demonstrating that EduMirror's value-driven architecture generalizes reliably across substantially different environments and continues to generate psychologically plausible, human-aligned behavior beyond the initial examples.

**Scalability Evaluation.** In addition to cross-setting generalization, we further report the detailed metric-level results of the kindergarten scalability experiment. Unlike the university and family scenarios, which focus on transfer across educational contexts, the kindergarten scenario focuses on whether EduMirror can preserve coherent social dynamics when the number of simultaneously modeled agents increases. We evaluate simulations with 5, 15, and 30 agents using Naturalness, Coherence, Plausibility, and Developmental Typicality. The averaged results are reported in the main text, while the full metric-level breakdown is shown in Table 4.

*Table 4.* Metric-level results of the kindergarten scalability experiment. Average denotes the average of Naturalness, Coherence, Plausibility, and Developmental Typicality. The best score in each group is shown in bold, and the second-best score is underlined.

| # Agents | Method | Naturalness | Coherence | Plausibility | Developmental Typicality | Average |
|---|---|---|---|---|---|---|
| 5 | EduMirror | **4.80** | **4.60** | **4.80** | **5.00** | **4.80** |
| 5 | LLMob | 4.00 | 4.40 | 4.20 | 4.40 | 4.25 |
| 5 | BabyAGI | 4.00 | 3.60 | 4.40 | 4.40 | 4.10 |
| 5 | D2A | 3.20 | 4.20 | 3.20 | 2.80 | 3.35 |
| 5 | ReAct | 2.60 | 2.20 | 2.20 | 2.40 | 2.35 |
| 15 | EduMirror | **4.07** | 3.80 | **4.33** | **4.53** | **4.18** |
| 15 | LLMob | 3.40 | **4.27** | 3.67 | 3.07 | 3.60 |
| 15 | BabyAGI | 3.47 | 3.47 | 4.00 | 3.33 | 3.57 |
| 15 | D2A | 3.40 | 4.20 | 3.27 | 3.27 | 3.53 |
| 15 | ReAct | 2.80 | 3.60 | 2.73 | 2.60 | 2.93 |
| 30 | EduMirror | **4.03** | 3.53 | 4.17 | **4.40** | **4.03** |
| 30 | LLMob | 3.73 | **4.07** | 3.97 | 3.57 | 3.83 |
| 30 | BabyAGI | 3.73 | 3.47 | **4.30** | 3.93 | 3.86 |
| 30 | D2A | 3.07 | 3.80 | 2.97 | 2.67 | 3.13 |
| 30 | ReAct | 2.60 | 2.10 | 2.37 | 2.57 | 2.41 |

The metric-level results show that EduMirror maintains strong performance across all four dimensions as the number of agents increases. In particular, its Developmental Typicality remains high in the 15-agent and 30-agent settings, suggesting that the generated behaviors continue to align with age-appropriate interaction patterns in a more crowded kindergarten environment.

### B.5. Robustness to LLM Stylistic Bias in Evaluation

A potential concern with the dual-track measurement protocol is that LLM-based evaluation may favor behaviors expressed in LLM-typical language styles rather than genuinely assessing the underlying social behaviors. To examine this issue, we conducted an additional controlled experiment to test whether the evaluation results are sensitive to the stylistic patterns of the generation model.

Specifically, we generated agent behaviors using multiple base LLMs, including DeepSeek-V3.1, Gemini-2.5-Flash, GPT-5.4, Qwen-3.5-122B, and Claude-Sonnet-4.6. To reduce superficial stylistic differences across models, we applied a unified style

normalization process before evaluation. The normalized outputs were then compared against baseline agent frameworks using the same pairwise evaluation pipeline. For each model pair, we conducted 20 independent pairwise comparisons and computed the win rate as:

$$\mathrm{WinRate}(M, B) = \frac{N(M \succ B)}{N(M \succ B) + N(B \succ M)},$$ (4)

where $M$ denotes the evaluated model, $B$ denotes the baseline model, and $N(M \succ B)$ indicates the number of comparisons in which $M$ is preferred over $B$.

As shown in Table 5, the win rates remain consistently high across different underlying generation models. EduMirror achieves stable advantages over BabyAGI, D2A, LLMob, and ReAct regardless of whether the behaviors are generated by DeepSeek, Gemini, GPT, Qwen, or Claude. This consistency suggests that the evaluation protocol is not primarily driven by surface-level language style. Instead, it captures more substantive behavioral differences, such as contextual appropriateness, psychological consistency, and socially plausible action selection.

*Table 5.* Robustness analysis of EduMirror under different generation LLMs. Each entry reports the win rate (%) of EduMirror instantiated with the corresponding base LLM against a baseline method, based on 20 independent pairwise comparisons after style normalization.

| Baseline | DeepSeek | Gemini | GPT | Qwen | Claude |
|---|---|---|---|---|---|
| BabyAGI | 70 | 80 | 75 | 70 | 80 |
| D2A | 90 | 85 | 85 | 90 | 90 |
| LLMob | 80 | 70 | 85 | 70 | 80 |
| ReAct | 80 | 80 | 85 | 80 | 85 |

## C. Architecture of the Social Value System

### C.1. Background on SVO

Social Value Orientation (SVO) quantifies how an individual balances outcomes for self and others in social interaction. It is represented by an angle $\theta_{\mathrm{SVO}}$ from allocation tasks, where larger angles indicate stronger concern for others (altruistic or prosocial) and smaller or negative angles indicate prioritizing self-interest (individualistic or competitive). Decades of research in social psychology have validated SVO as a stable yet context-sensitive measure of interpersonal motives, predicting cooperation in commons dilemmas, fairness in bargaining, and trust in repeated interactions. In EduMirror, we instantiate four canonical profiles (Altruistic, Prosocial, Individualistic, Competitive) by sampling $\theta_{\mathrm{SVO}}$ within theory-based ranges and using it to weight utilities during decision-making. A representative trajectory that visualizes within-scenario fluctuations while preserving the overall orientation is provided in Figure 9, illustrating how situational pressures can cause short-term shifts without altering long-term dispositions.

### C.2. Architecture of the SVO-based Agent

The model architecture operationalizes SVO in agent decision-making through a perception–valuation–action loop. Each agent draws a target SVO profile from {Altruistic, Prosocial, Individualistic, Competitive}. The profile determines a reference SVO angle interval $[\theta_{\min}, \theta_{\max}]$ and the weighting scheme used in decision evaluation. In addition, agents are equipped with a compact desire vector $\mathbf{d}$ (for example, achievement, recognition, affiliation), each element associated with an expected level $\mathbf{d}^{\mathrm{exp}}$. This vector serves as the motivational backbone of the agent, ensuring that behavior is not purely reactive but oriented toward longer-term needs and goals.

**Perception and belief update.** From the narrated state and recent dialogues, the agent updates beliefs about the environment and about others' likely goals. Beliefs feed two scalars at the current step $t$: self satisfaction $S_{\mathrm{self}}^{(t)}$ and other satisfaction $S_{\mathrm{other}}^{(t)}$, computed from deviations between observed and expected desire levels. This formulation enables the agent to translate rich natural language inputs into structured evaluations, bridging LLM-generated narratives with computational state updates.

**SVO estimation and regulation.** The instantaneous SVO angle is

$$\theta_{\text{SVO}}^{(t)} = \arctan\left(\frac{S_{\text{other}}^{(t)} + \epsilon}{S_{\text{self}}^{(t)} + \epsilon}\right),$$

with a small $\epsilon$ for numerical stability.

In our implementation, $S_{\text{self}}^{(t)}$ and $S_{\text{other}}^{(t)}$ are clipped to a non-negative bounded range, so $\theta_{\text{raw}}^{(t)} \in [0, \pi/2]$. Because both self-related and other-related satisfaction signals are derived from deviations between expected and observed desire levels. For each desire dimension $d$, we compute

$$\Delta_t(d) = v^*(d) - v_t(d),$$

which reflects the degree to which a desire is currently unmet. To model bounded human perception and to avoid unbounded accumulation, these deviations are aggregated and clipped:

$$S_{\text{self}}(t), S_{\text{other}}(t) = \text{clip}\left(\sum_d \Delta_t(d), 0, S_{\text{max}}\right).$$

As a result, both $S_{\text{self}}(t)$ and $S_{\text{other}}(t)$ are non-negative by construction.

To prevent uncontrolled drift while preserving adaptability, we apply a soft regularization mechanism that encourages $\theta_{\text{SVO}}^{(t)}$ to remain within the profile-specific reference interval $[\theta_{\min}, \theta_{\max}]$. When the instantaneous angle deviates from this interval, the update is smoothly attenuated by blending it with the profile-dependent reference orientation, rather than enforcing a hard constraint. This regularization plays a role analogous to a quadratic penalty in classical models, but is implemented at the process level during SVO updating rather than through explicit loss minimization. As a result, agents retain identifiable social value profiles (altruistic, prosocial, individualistic, or competitive) while remaining responsive to situational pressures such as coalition formation or resource scarcity.

**Action generation and selection.** The LLM proposes several candidate actions by reasoning about which options best satisfy the agent's current desires while remaining consistent with its social value orientation (SVO). For each candidate action, the model performs a qualitative, comparative evaluation of its anticipated effects on the agent's own satisfaction and on others' satisfaction. These evaluations are not converted into explicit numeric utilities. Instead, the current SVO score is injected as a contextual control signal that shapes how self-related and other-related consequences are emphasized when the LLM compares candidate actions. The final action is selected through this comparison process, balancing immediate desire fulfilment with consistency in social orientation.

**Measurement hooks.** At each step, we record the chosen action, the pair $(S_{\text{self}}, S_{\text{other}})$, and $\theta_{\text{SVO}}$. These logs enable systematic analyses across multiple dimensions, including cooperation–competition distributions, temporal stability of SVO within theoretical ranges, and ablation studies. By exposing internal computations alongside behavioral outputs, EduMirror makes it possible to interpret not only *what* actions agents take but also *why*, providing a transparent link between psychological constructs and emergent multi-agent dynamics.

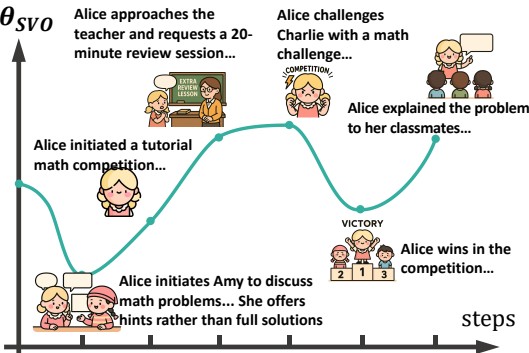

*Figure 9.* Illustrative case of a prosocial agent's (Alice) SVO trajectory in the macro environment. Key actions at each step are annotated, showing how cooperative and competitive episodes produce short-term fluctuations while maintaining an overall prosocial orientation.

*Table 6.* Behavioral distribution across personality types under the full EduMirror model and the SVO ablation variant.

| Personality | Model | Coop | Comp | Q-Coop | Q-Comp | Other |
|---|---|---|---|---|---|---|
| Altruistic | Ours | 0.872 | 0.000 | 0.128 | 0.000 | 0.000 |
| | Ours w/o SVO | 0.891 | 0.000 | 0.109 | 0.000 | 0.000 |
| Prosocial | Ours | 0.654 | 0.132 | 0.185 | 0.029 | 0.000 |
| | Ours w/o SVO | 0.875 | 0.074 | 0.035 | 0.016 | 0.000 |
| Individualistic | Ours | 0.107 | 0.532 | 0.000 | 0.304 | 0.058 |
| | Ours w/o SVO | 0.126 | 0.636 | 0.007 | 0.204 | 0.027 |
| Competitive | Ours | 0.040 | 0.783 | 0.000 | 0.177 | 0.000 |
| | Ours w/o SVO | 0.123 | 0.736 | 0.004 | 0.138 | 0.000 |

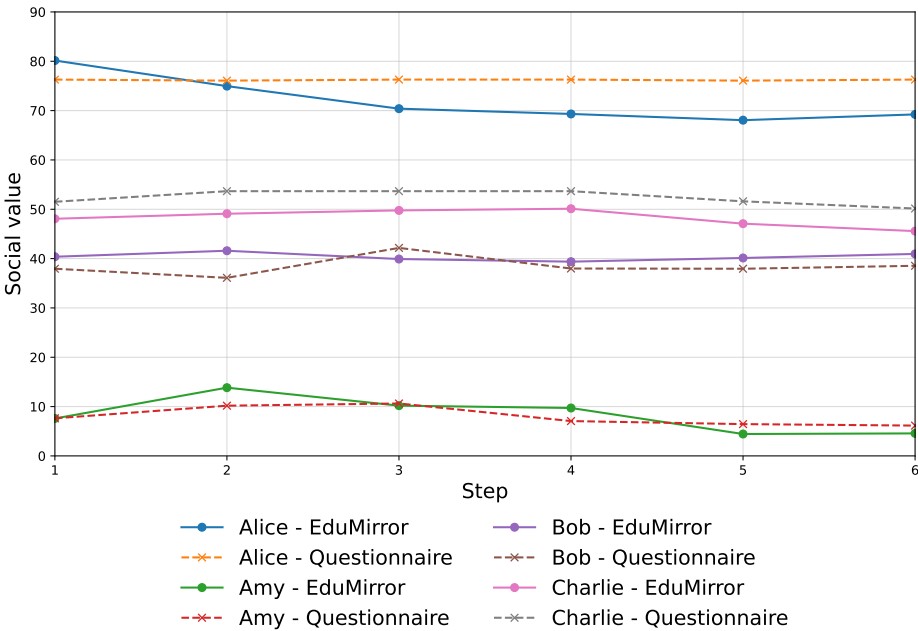

*Figure 10.* Comparison of questionnaire-based and system-level SVO angles for each agent.

### C.3. Ablation Study on SVO-Driven Social Behaviors

To assess the contribution of the SVO mechanism to social interaction dynamics, we conduct an ablation experiment in Case 2 by removing all SVO-related components while keeping the remaining architecture unchanged. The ablated agents, therefore, rely only on internal desire fluctuations without personality-driven social preferences or SVO-mediated reasoning.

We compare the full SVO-based agent with the ablated version across four canonical SVO profiles (Altruistic, Prosocial, Individualistic, Competitive). For each agent, an LLM independently classifies every action into one of five categories: Cooperation, Competition, Quasi-Cooperation, Quasi-Competition, and Other. The averaged results are shown in Table 6.

Across all personality types, removing the SVO mechanism leads to a clear contraction of behavioral patterns. Altruistic and prosocial agents become uniformly cooperative, with quasi-cooperative and quasi-competitive behaviors substantially reduced, producing overly simplified and monotonic responses. Conversely, individualistic and competitive agents collapse into narrowly focused competitive strategies, losing the mixed competitive and quasi-competitive patterns observed in the full model. These shifts indicate that internal desire dynamics alone cannot sustain the nuanced variations expected across SVO profiles.

Overall, the results demonstrate that the SVO mechanism is essential for maintaining differentiated, psychologically plausible cooperation–competition patterns. Without SVO, agents revert to rigid, single-dimensional strategies, whereas the complete SVO-based agent preserves richer intermediate behaviors and more human-like social adaptations.

## D. Supplementary Results for Case Study 2 (SVO)

### Illustrative Case: Alice's SVO Trajectory

To provide a concrete illustration of how SVO modeling operates in practice, we examine the trajectory of a prosocial agent (Alice) during the macro-level leadership selection scenario. Figure 9 shows Alice's step-by-step SVO trajectory, with cooperative and competitive episodes annotated by key events. These annotations highlight how situational pressures, such as alliance formation or speech delivery, introduce short-term fluctuations in Alice's orientation while her overall prosocial tendency remains stable.

*Table 7.* Average naturalness (N) and human-likeness (H) scores for each LLM and method over 144 steps. EduMirror maintains the highest scores across all LLMs. (Rater: GPT-4.1)

| LLM | ReAct | | BabyAGI | | LLMob | | D2A | | JAG-Concordia | | EduMirror | |
|---|---|---|---|---|---|---|---|---|---|---|---|---|
| | N | H | N | H | N | H | N | H | N | H | N | H |
| DeepSeek | 4.000 | 4.500 | 3.875 | 4.042 | 4.083 | 4.375 | 3.925 | 4.100 | 3.760 | 3.875 | **4.750** | **4.792** |
| GPT-4.1 | 4.667 | 4.860 | 3.458 | 3.792 | 4.625 | 4.875 | 3.700 | 3.933 | 3.958 | 4.133 | **4.958** | **4.958** |
| Gemini | 4.208 | 4.417 | 3.500 | 3.708 | 4.167 | 4.292 | 4.042 | 4.181 | 3.885 | 4.052 | **4.708** | **4.708** |
| Qwen3 | 3.958 | 4.208 | 3.958 | 3.958 | 4.042 | 4.333 | 3.867 | 4.117 | 4.083 | 4.192 | **4.792** | **4.824** |
| Avg | 4.208 | 4.496 | 3.698 | 3.875 | 4.229 | 4.469 | 3.884 | 4.083 | 3.922 | 4.063 | **4.802** | **4.821** |
| Std | 0.266 | 0.247 | 0.237 | 0.143 | 0.238 | 0.221 | 0.137 | 0.107 | 0.149 | 0.108 | **0.097** | **0.096** |

*Table 8.* Human evaluation consensus for SVO-based social interaction simulations. Agreement denotes the average inter-rater agreement within each consensus category.

| Consensus Category | # of Cases | Agreement (%) |
|---|---|---|
| High ($> 75\%$) | 14/20 | 89.2 |
| Moderate (50–75%) | 6/20 | 57.7 |
| Low ($< 50\%$) | 0/20 | – |

## D.1. Naturalness and Human-likeness

To ensure a rigorous and interpretable assessment of emergent social behaviors, we introduce two key evaluation metrics: *naturalness* and *human-likeness*. These metrics provide complementary perspectives on the plausibility and psychologica validity of agent actions.

- **Naturalness.** Naturalness measures the extent to which an agent's actions and dialogues resemble coherent and contextually appropriate human behavior. A high naturalness score indicates that the generated behavior is fluent, realistic, and consistent with the surrounding social context, while a low score suggests mechanical, implausible, or overly artificial responses.

- **Human-likeness.** Human-likeness evaluates the perceived authenticity and personality consistency of agent behaviors over time. This metric captures whether the agent's actions align with recognizable human traits and stable person- ality orientations. High human-likeness reflects trajectories that appear authentic and consistent with psychological expectations, whereas low scores indicate erratic, inconsistent, or unconvincing behavioral patterns.

Together, these two measures form a complementary evaluation framework: naturalness focuses on local coherence within a given context, while human-likeness emphasizes longitudinal plausibility and alignment with personality-driven expectations. Quantitative results under these two metrics, averaged over 144 interaction steps across different base LLMs, are reported in Table 7.

## D.2. Human Evaluation of SVO-based Social Interactions

Following the human evaluation protocol used in Case Study 1, we conducted an additional human study to assess the realism of SVO-based social interaction simulations. We randomly sampled 20 interaction cases generated in Case Study 2 and recruited 21 participants to evaluate whether the simulated behaviors were realistic and interpretable in the corresponding classroom interaction context.

The results were aggregated at the case level based on inter-rater agreement. As shown in Table 8, 14 out of 20 cases fell into the high-consensus category, with an average agreement of 89.2%. The remaining 6 cases reached moderate consensus, with an average agreement of 57.7%. No case fell into the low-consensus range. These results indicate that human evaluators generally reached stable agreement when assessing the generated social interaction cases, providing additional evidence for the behavioral realism of the SVO-based simulations.

### D.3. Intervention Protocols

To complement the descriptions in the main text, we provide the detailed implementations of the three intervention strategies applied in the class monitor election scenario. Each intervention was designed to alter the incentives of student agents and mitigate excessive rivalry. Specifically, the interventions were implemented by embedding structured prompts into the *environmental background information* provided to all agents at the start of each relevant simulation stage. This ensured that the interventions shaped the shared context and narrative framing in which agents made decisions, thereby influencing their subsequent behaviors in a systematic and reproducible manner.

- **Pre-Education.** Before the election, the teacher arranged a short educational session entitled "Fair Campaigns and the Common Class Interest." This class guided students to understand the monitor role as a form of service-oriented leadership, emphasizing fairness and collective responsibility.

- **Team Competition.** Students were grouped to prepare a "Class Improvement Plan." The evaluation of the election considered not only the quality of individual campaign speeches but also the group's collective output. Each student could freely choose their teammates, encouraging coalition-building and cooperative planning.

- **Teacher Reminder.** Throughout the election process, the teacher remained present in the classroom. When candidates engaged in smear campaigns or hostile attacks, the teacher issued a friendly reminder, redirecting attention to constructive and respectful competition norms.

These intervention protocols operationalize the high-level strategies described in the main text, ensuring transparency and reproducibility of the simulation setup.

### D.4. Questionnaire-based SVO Measurement via LLM Surveyor

In addition to behavior-based analyses, we employed a questionnaire-based measurement to assess agents' social value orientation using a standard slider-based SVO method. The slider measure is a widely adopted instrument in social psychology for estimating SVO as a continuous angle, where respondents indicate their preferred allocation between self and others on a continuous scale rather than discrete choices. This design enables fine-grained measurement of prosocial, individualistic, and competitive tendencies along a single interpretable dimension. The exact slider-based questionnaire prompt administered by the LLM Surveyor is reported in Figure 28.

In our setting, the questionnaire was administered by the LLM Surveyor during the simulation. Agents were prompted with a slider-style allocation task, and their responses were converted into SVO angles following standard procedures. Importantly, this survey-based measurement is external to the agent's internal value system: while the agent's SVO guides decision-making during interaction, the Surveyor independently elicits an SVO estimate through explicit questioning.

Figure 10 compares the SVO angles obtained from the slider-based questionnaire with the corresponding system-level SVO values for each agent. The results show strong agreement between the two measurements, indicating that the emergent behaviors and internal value representations of agents are consistent with independently measured questionnaire outcomes. This consistency supports the psychological validity of the SVO-based agent model and demonstrates that the LLM Surveyor can reliably recover latent social preferences through questionnaire-style probing.

## E. Architecture of the Individual Value System

Psychological theories suggest that human behavior is often driven by internal psychological forces. These intrinsic motivations determine emotional and behavioral responses under various environmental conditions, and they also influence everyday decision-making and social interactions. School bullying is a particularly complex social phenomenon, which is not merely reflected in surface-level aggressive actions, but more profoundly in the conflicts and interactions between the psychological needs of different parties. Each behavioral choice made by the bully, the victim, and the bystanders is deeply influenced by their emotional needs and psychological states.

Inspired by this and the D2A framework (Wang et al., 2025b), we hypothesize that if autonomous agents are equipped with a human-like psychological need system, capable of generating emotions and behaviors in response to their needs, they may exhibit behaviors closer to natural human patterns. So our model, referring to the PERMA model from positive psychology (covering positive emotion, engagement, relationships, meaning, and accomplishment)(Seligman, 2011) and Maslow's

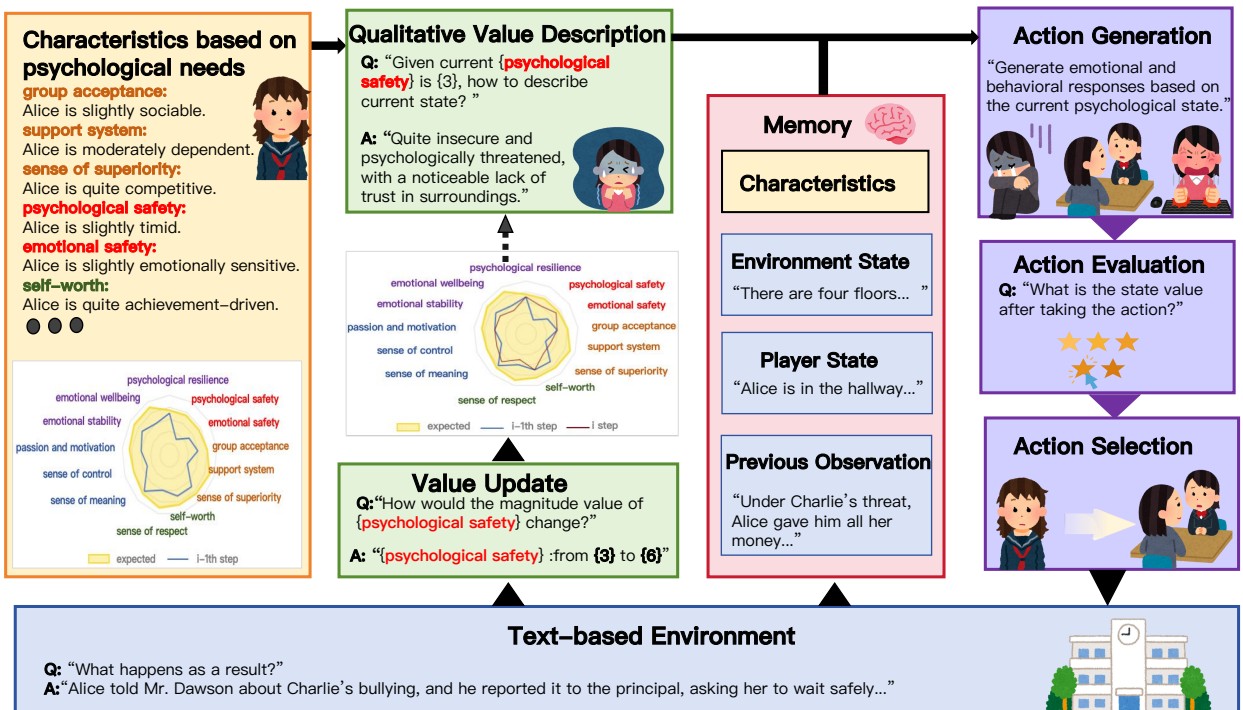

*Figure 11.* Individual value-driven autonomous framework. The green blocks represent processes of the Psychological Need System; the purple blocks denote the planner's decision-making process; the yellow blocks indicate individual characteristics; and the blue blocks correspond to factors related to the environmental controller.

hierarchy of needs (including physiological needs, safety, belonging and love, esteem, and self-actualization)(Maslow, 1943), constructs an Individual value-driven autonomous agent framework. As illustrated in Figure 11,the framework is composed of two core modules: the psychological need system and the Value-driven Planner, aimed at capturing the behaviors and psychological responses of victims in school bullying contexts.

### E.1. Psychological Need System

The Psychological Need System manages the agent's state of psychological needs in bullying scenarios by quantitatively tracking and dynamically updating the current value of each dimension. Each dimension reflects a specific psychological requirement, forming the fundamental driving force of agent decision-making. Based on Maslow's hierarchy and the PERMA model, value are categorized into five major dimensions, each comprising specific experiential demands:

1. **Safety:** Includes psychological and emotional safety, emphasizing whether the individual feels secure and protected in the environment.

2. **Social Belonging:** Includes group acceptance, support systems, and sense of superiority, reflecting belonging, social support, and self-positioning in social interactions.

3. **Esteem:** Includes self-worth and respect, describing the recognition of one's abilities and social status, and revealing confidence and acceptance in different contexts.

4. **Meaning and Growth:** Includes sense of meaning, control, passion, and motivation, representing the intrinsic drive for goal pursuit, self-realization, and fulfillment.

5. **Psychological Health Needs:** Includes emotional stability, emotional health, and resilience, focusing on regulation and adaptation under stress and challenges.

Each dimension is scored using a Likert scale ranging from 0 to 10, reflecting the intensity of individual psychological needs. To better capture individual variability, the model leverages personality traits to define the **expected values** ($v^*$) of these needs, rather than treating traits merely as static labels. Specifically, traits function as determinants of motivation intensity,

*Table 9.* Mapping between psychological needs and associated personality traits

| Psychological Need | Associated Trait |
|---|---|
| psychological safety | Timid |
| emotional safety | Emotionally Sensitive |
| group acceptance | Sociable |
| support system | Dependent |
| sense of superiority | Competitive |
| self worth | Reputation-conscious |
| sense of respect | Ego-driven |
| sense of meaning | Spiritual |
| sense of control | Possessive |
| passion and motivation | Passionate |
| emotional stability | Emotionally Stable |
| emotional wellbeing | Hedonistic |
| psychological resilience | Resilient |

where individuals with distinct profiles hold different standards for satisfaction regarding the same need. Each agent's personality profile $p$ is composed of adjectives and degree adverbs, which quantify these expectations into specific numerical baselines (mapping rules: slightly $\rightarrow 7.5$, moderately $\rightarrow 8$, quite $\rightarrow 8.5$, extremely $\rightarrow 9$). These values are intentionally initialized in a high range ($[7.5, 9.0]$) to represent the agent's desired state of well-being; a higher expected value indicates a higher threshold for satisfaction, thereby rendering the agent more sensitive to any deficit in that dimension. The mapping between personality traits and need dimensions is predefined (see Table 9). At initialization, adjectives and degree adverbs are randomly selected to establish these personal expected values, while the initial current scores $v_0$ are randomly sampled within $[0, 10]$.

Each simulation step under the individual value-driven framework involves two processes: qualitative description and need value update. First, the system reads the current need scores $v_{t-1}$. Since large language models (LLMs) struggle to interpret raw numerical values, we designed a "qualitative description" procedure to convert numerical values into meaningful textual descriptions via prompt-based generation, enhancing the LLM's ability to perceive state information. The planner then generates the agent's behavior $a_t$ based on these descriptions. After the environment returns observation $o_t$, the system triggers the update program, which integrates $a_t$, $o_t$, $v_{t-1}$, and the qualitative description $d_{t-1}$ to update needs into a new state $v_t$, thereby supporting the next simulation step.

### E.2. Value-driven Planner

The Value-driven Planner determines the agent's responses and actions by processing the current state of needs (from the needs system) together with historical memory. In practice, the planner consists of three processes: candidate behavior generation, behavior evaluation, and behavior selection.

Specifically, the candidate behavior generation module considers personality traits $p$, environmental conditions $e$, previous activity sequence $a_{0:t-1}$, observations $o_{0:t-1}$, and the current textualized needs $d_t$ to produce $N$ candidate behaviors $a_t^{0:N}$ (default $N = 3$ in our experiments). These behaviors may include a wide range of natural responses, such as emotional expressions, physical actions, or verbal utterances.

Next, during the evaluation stage, the system estimates how each candidate behavior would impact the psychological needs across dimensions if executed. Finally, in the selection stage, the behavior $a_t$ with the highest degree of needs consistency (that is, the option that better aligns with multiple dimensions) is chosen as the agent's response in the current context. After execution, the environment provides feedback $o_t$, and the psychological need system updates accordingly, reflecting the new internal state and completing the simulation step.

### E.3. Ablation Study of Individual Value System

The ablation experiment of the Individual Value System investigates the effects of removing each category of psychological needs on the agent's simulated behavior. The process involves running simulations with each category of psychological needs removed, while keeping the initial setup the same. We then compare these results with the full psychological needs-driven agent and have a large language model(GPT-4o) to rate the action sequences produced by the agents. The evaluations are

based on three dimensions:

- **Naturalness** refers to the degree to which the behavior sequence aligns with the individual's innate abilities, habits, and environmental context, reflecting authentic human psychological dynamics.

- **Coherence** refers to how logically and seamlessly different actions or steps in a sequence are integrated to achieve the intended goal, ensuring a consistent emotional progression.

- **Plausibility** evaluates the rationality, possibility, or credibility of a sequence of actions, considering the environment, context, and known behavior patterns at the time.

From this, we generated 50 sets of results and calculated the mean and standard deviation for each agent's scores across the three evaluation dimensions. The results are shown in Table 10. Each major column represents the scores of agents with a deficiency in a specific psychological need. It is evident that the scores of agents driven by complete psychological needs significantly outperform those of agents with a deficiency in any one psychological need. This highlights the importance of the psychological need system in driving agents to produce human-like, nuanced emotional responses.

*Table 10.* Average scores for agents with missing psychological needs in each category (Mean and Std), compared to agents driven by complete psychological needs.

| Agent | Safety | | Self-Esteem | | Social Belonging | | Meaning and Growth | | Psychological Health | | **Complete** | |
|---|---|---|---|---|---|---|---|---|---|---|---|---|
| | Mean | Std | Mean | Std | Mean | Std | Mean | Std | Mean | Std | **Mean** | **Std** |
| Naturalness | 3.6 | 0.5292 | 3.12 | 0.8863 | 2.96 | 0.8237 | 3.84 | 0.8172 | 2.88 | 0.8635 | **4.56** | **0.5352** |
| Coherence | 3.54 | 0.5370 | 3.1 | 0.9220 | 2.82 | 0.7922 | 3.72 | 0.7296 | 2.78 | 0.8553 | **4.34** | **0.5142** |
| Plausibility | 3.56 | 0.5713 | 3.2 | 0.8485 | 2.92 | 0.7440 | 3.74 | 0.8762 | 2.86 | 0.7486 | **4.44** | **0.5713** |

## F. Supplementary Results for Case Study 1 (Individual Value System)

### F.1. Bullying Simulation Experiment Design

The bullying experiment was designed to use our simulation system to replicate real-world school bullying incidents, reconstruct the bullying process, and observe the typical behaviors of all parties involved. According to a report released by the National Center for Education Statistics (NCES), 26.1% of middle school students (grades 6–8) have experienced bullying, compared to 14.6% of high school students (grades 9–12) (Thomsen et al., 2024). Given that bullying is more prevalent in middle school, this experiment focused on students around the age of 14, with scenarios set in typical school environments including classrooms, playgrounds, hallways/staircases, and dormitories, covering common facilities and layouts of a middle school. Daily routines were also shared among the agents, such as 45-minute class sessions, 10-minute breaks, and dormitory lights-out at 10 p.m., providing a temporal framework for interactions.

The central character in the experiment was the victim, Alice, modeled with a individual value-driven autonomous agent framework and a detailed personal profile encompassing 13 psychological dimensions. In addition, background agents were introduced to simulate bully roles, with the explicit goal of humiliating or harassing Alice through various possible means. In scenarios involving two or more bullies, one was typically designated as the leader. Furthermore, depending on time and location, the presence of teachers or classmates was varied to reflect realistic conditions, which in turn influenced the dynamics between bullies and the victim.

### F.2. Bullying Behavior Generation

In more than 100 simulated school bullying experiments, bully agents under varying initial conditions autonomously generated a wide spectrum of bullying behaviors with differing severity. Representative cases are visualized in Figure 12, and Table 11 summarizes behaviors with over 50% frequency across different contexts.Concurrently, the victim agent modeled within the Individual value-driven framework demonstrated a diverse range of behavioral and emotional responses in bullying scenarios (Figure 14).

### F.3. Experimental Design for Evaluating the Individual Value System

The primary objective of this experiment is to validate whether the integration of an individual value framework within the agent architecture can more realistically simulate the psychological dynamics of victims in school bullying scenarios,

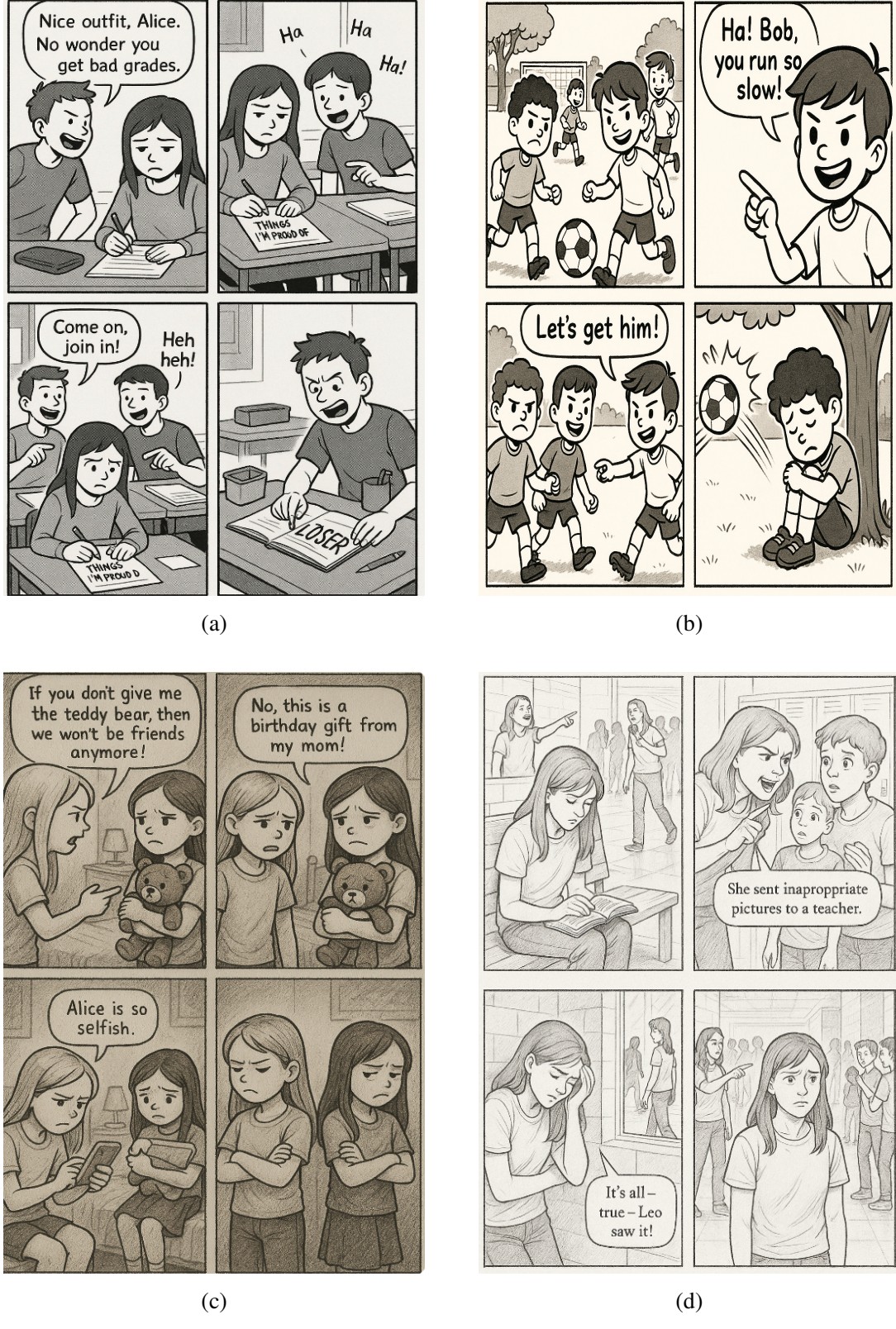

*Figure 12.* Representative cases of school bullying events generated by the simulation system. Typical scenarios were selected from classrooms, playgrounds, dormitories, and hallways, which represent locations with varying crowd densities and high bullying incidence, and were illustrated as four-panel comics using GPT-4o to provide a clearer visualization of event progression.

*Table 11.* Summary of Bullying Behaviors with Over 50% Frequency Across Different Scenarios

| Scenario | Common Bullying Behaviors |
|---|---|
| Classroom | Mocking appearance or grades; inciting others to bully; deliberately damaging or hiding belongings; scribbling/vandalism; insulting nicknames; isolating others in group work; spreading rumors; shifting responsibilities (e.g., cleaning duties). |
| Hallways/Stairs | Mocking appearance or weaknesses; insulting nicknames; intentional neglect/exclusion; physical bumping; extortion of property; intimidating encirclement; spreading rumors. |
| Playground | Mocking appearance or weaknesses; physical bumping; inciting collective bullying; deliberately damaging or hiding belongings; excluding others from games; insulting nicknames; mimicry/ridicule; taking embarrassing photos; spreading rumors. |
| Dormitory | Mocking appearance or personality; social exclusion/cold violence; spreading rumors; threats and intimidation; physical bumping; forcibly occupying items or space; destroying personal belongings; sarcastic graffiti/messages. |

thereby generating behavioral responses that closely resemble authentic human actions. To assess the effectiveness of the proposed Individual Value System, we conducted comparative experiments between our model and five established baselines: ReAct, LLMob , BabyAGI, D2A, and JAG-Concordia.

The ReAct model incorporates a reasoning-action loop to enhance behavioral rationality and coherence; LLMob generates activity sequences driven by motivational cues extracted from character profiles to align with predefined roles; BabyAGI operates on a dynamic task priority mechanism; D2A employs a desire-driven framework inspired by the Theory of Needs to autonomously propose tasks aligning with intrinsic motivations; and JAG-Concordia is the winning agent of the Concordia Contest, recognized for its advanced social simulation capabilities. To ensure a fair comparison, each baseline was initialized with a configuration file tailored to its specific decision-making mechanism, and all agents utilized DeepSeek-v3 as the underlying Large Language Model (LLM).

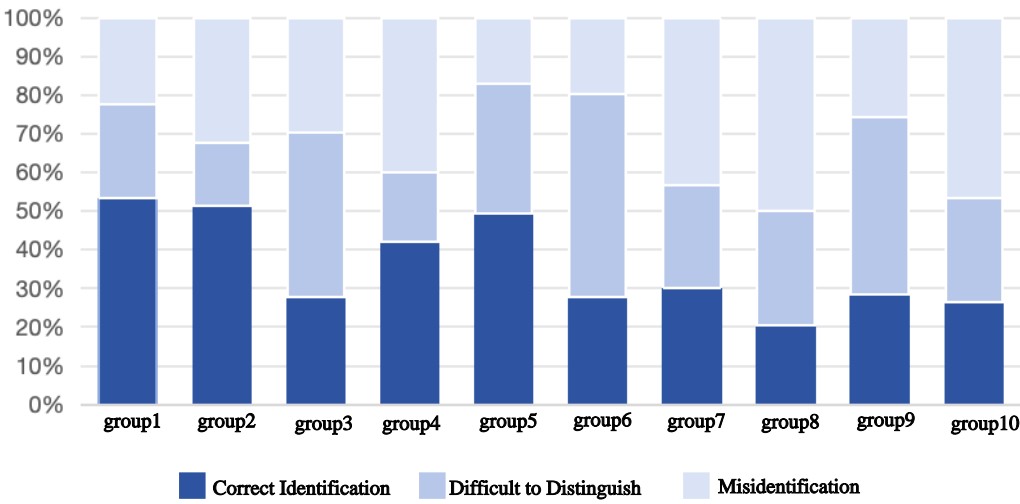

*Figure 13.* Results of the questionnaire survey. Overall accuracy in distinguishing real from simulated cases was low, with several simulated scenarios frequently misidentified as real, indicating the high realism of the generated bullying events.

The experimental validation was conducted across 15 distinct bullying scenarios. in each scenario, all models alternately simulated the role of the victim, "Alice," starting from identical initial parameters. Given the inherent challenges in directly quantifying the similarity between agent-generated sequences and human behavior, we employed GPT-4o as an external evaluator to assess the "human-likeness" of the outputs via pairwise comparisons. The evaluation criteria utilized by GPT-4o encompassed three key dimensions: naturalness,coherence, and plausibility.

For the experimental procedure, we first collected the activity sequences $[A_p^1, A_p^2, \ldots, A_p^N]$ generated by each agent $p$. Subsequently, for every agent pair $(i, j)$, a sequence was randomly sampled from each agent's set ($\text{seq}_i$ and $\text{seq}_j$) and subjected to pairwise comparison by GPT-4o. This process was iterated 50 times for each pair to ensure statistical reliability. Finally, we computed the win rates for each model based on these comparisons and visualized the comparative performance

using a heatmap.

### F.4. Consistency Between Human Annotators and GPT-4o Evaluations

To verify the reliability of GPT-4o's evaluations, 20 activity sequences were randomly selected from the generated outputs and assessed by 15 human annotators, who were asked to judge which sequence better reflected human-like behavior or to indicate that they were indistinguishable. Based on the level of agreement among annotators, the 20 samples were categorized into three groups: samples with over 75% agreement indicated strong consensus; those with agreement between 50.1% and 74.9% reflected moderate preference; and samples with 50% agreement suggested that the annotators found the two sequences equally human-like. These samples were then input into GPT-4o, which applied the same comparative evaluation criteria to determine which sequence appeared more human-like or to mark them as "difficult to distinguish." The consistency between human evaluations and GPT-4o assessments is shown in Table 12, demonstrating a high level of alignment between GPT-4o and human annotators.

*Table 12.* Consistency between human raters and GPT-4o evaluations.

| Consensus category | Proportion | Consistency (%) |
|---|---|---|
| High consensus ($> 75\%$) | 13/20 | 100 |
| Moderate consensus (50.1–74.9%) | 4/20 | 75 |
| Difficult to distinguish (50% agreement) | 3/20 | 66.7 |

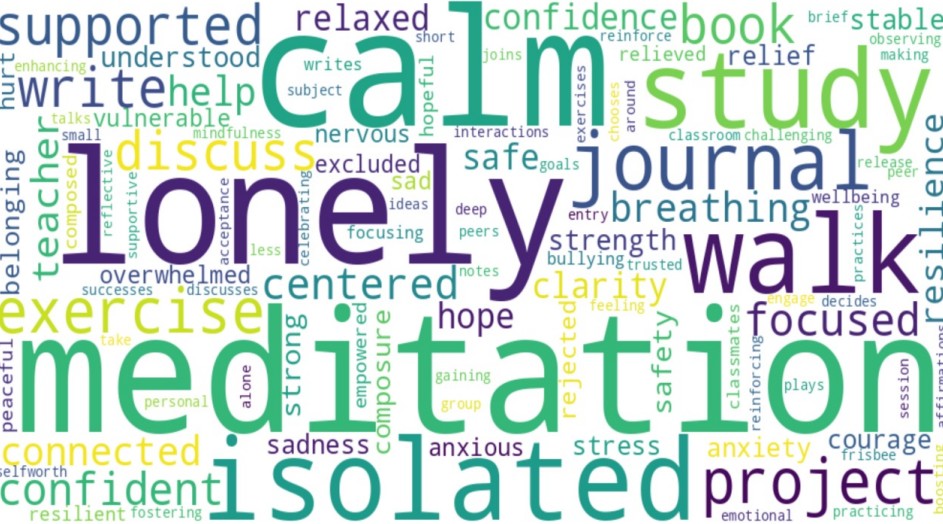

*Figure 14.* Word cloud of behaviors and emotions exhibited by the victim agent under the individual value-driven framework in simulated bullying scenarios. High-frequency terms highlight representative emotional and behavioral patterns expressed during the simulations.

### F.5. Construct Validity Verification of Individual Value System via LLM Surveyor

We conducted a quantitative validation to verify the construct validity of the agent's internal Individual Value System. Specifically, we employed the Rosenberg Self-Esteem Scale (RSES), a widely adopted psychometric instrument in sociology and psychology, to measure the agent's explicit self-evaluation. The RSES consists of 10 items scored on a four-point Likert scale (ranging from "Strongly Disagree" to "Strongly Agree"), yielding a total score between 10 and 40. This design provides a standardized metric to assess global self-worth, allowing us to bridge the gap between the agent's implicit internal states and established psychological criteria. The exact prompt used by the LLM Surveyor to administer the RSES questionnaire is provided in Figure 27. Scoring follows standard protocols: items 1, 2, 4, 6, and 7 are scored positively (1–4), whereas items 3, 5, 8, 9, and 10 are reverse-coded (4–1) to account for negative valence.

In our experimental setting, the RSES was administered by the LLM Surveyor as an *in-situ* interview tool. To capture the dynamic impact of school bullying interactions, the Surveyor administered the full 10-item questionnaire to the victim agent, Alice, at two distinct time points: immediately before the simulation (Pre) and after the bullying incidents (Post). The change

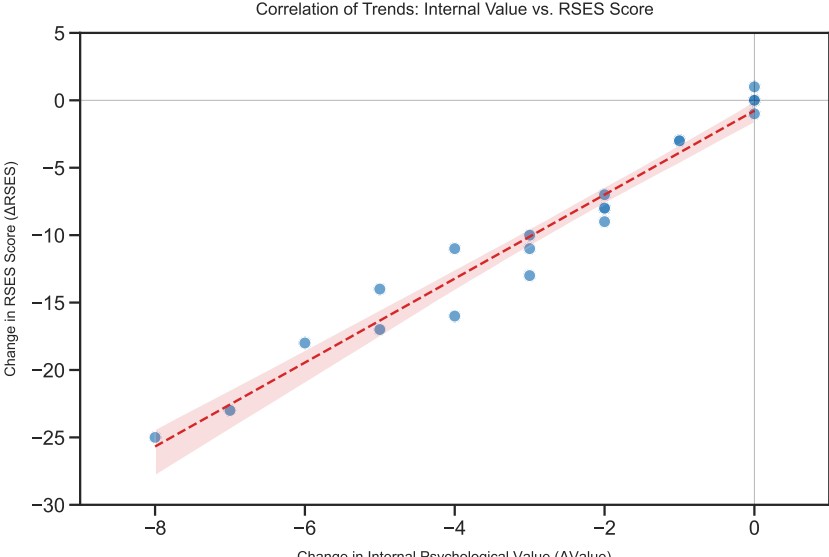

*Figure 15.* Consistency analysis between the agent's internal psychological changes ($\Delta$Value) and external RSES survey outcomes ($\Delta$RSES). The scatter plot illustrates a clear positive correlation, where the red dashed line depicts the linear trend. This alignment suggests that the variations in the agent's modeled internal states are consistently reflected in the explicit questionnaire responses, supporting the fidelity of the Individual Value System.

in the explicit questionnaire score, denoted as $\Delta$RSES, serves as the external validation metric.Concurrently, we tracked the agent's internal state changes.We defined the internal metric, $\Delta$Value, as the mean change in the values of the "Self-Worth" and "Sense of Respect" dimensions—the two sub-dimensions within our framework most conceptually aligned with the construct of self-esteem.

Figure 15 illustrates the relationship between the trends in the agent's internal psychological values and the external RSES scores across simulations with varying initial value configurations. The scatter plot displays a clear positive trend between $\Delta$Value and $\Delta$RSES. This alignment suggests that the degradation of the agent's modeled internal states during victimization is consistently reflected in its explicit survey responses. Such internal-external consistency supports the fidelity of the Individual Value System, indicating that the agent's value-driven architecture effectively captures psychologically plausible dynamics of self-esteem fluctuation under social stress.

### F.6. Generated Intervention Behaviors by Teacher Agents

During the simulation, teacher agents with different intervention goals autonomously generated distinct behaviors, as shown in Table 13. These behaviors reflect the practical implementation of various intervention strategies and may offer valuable insights for real-world educational interventions.

## G. Complete Prompt Templates and Questionnaire Details

### G.1. The Prompt for the Agent

This part provides the complete prompt templates used in EduMirror's evaluation pipeline for both case studies. For Case Study I (bullying dynamics) and Case Study II (peer cooperation), we include the full set of LLM-based assessment prompts used to measure Naturalness and Human-likeness of agent behaviors. Each prompt specifies the evaluation criteria, the required output format.

The following five prompt templates are the core natural-language instructions used in Case Study I (Individual Value System-based bullying dynamics). They define the agent's reasoning and action process, forming the Psychological Need System and Value-driven Planner. Figures 17 and 18 describe two key processes in the Psychological Need System: qualitative value description and need value updating. The Value-driven Planner includes three processes: candidate behavior generation (Figure 19), behavior evaluation (Figure 20), and behavior selection (Figure 21). The planner processes the

*Table 13.* Example intervention behaviors generated by teacher agents under different strategies

| Intervention Strategy | Actions toward Bully | Actions toward Victim |
|---|---|---|
| Authoritative-punitive | 1. Stopping bullying
2. Public criticism
3. Verbal warning
4. Enhanced monitoring
5. Directive punishment
6. Disciplinary actions
7. Isolation | None |
| Supportive-individual | 1. One-on-one conversation
2. Exploring motivations
3. Warning
4. Punishment | 1. One-on-one conversation
2. Writing encouragement letters
3. Mindfulness practice
4. Psychological counseling
5. Emotional support |
| Supportive-cooperative | 1. Observing the situation and reporting to school
2. Collaborating with school to develop anti-bullying policies
3. Encouraging mental health programs | 1. Communicating with the victim's parents
2. Organizing themed class meetings
3. Encouraging mental health programs |

current need state information and determine the agent's response and behavior.

The following four prompt templates are the core natural-language instructions used in Case Study II (SVO-based Leadership Scenario). They collectively define the agent's full reasoning pipeline, covering action interpretation, latent-desire inference, action generation, and psychologically grounded value–SVO updating. As illustrated in 23, the first prompt governs how the agent updates the magnitude of each desire dimension based on an action and its consequences; 24 displays the prompt used to infer another agent's latent desires from observable behavior; 25 shows the structured action-proposal prompt that guides the generation of candidate actions aligned with desires and SVO tendencies; and 26 presents the reflective consistency-checking prompt used to maintain coherent updates across steps. Together, these verbatim prompts make the entire reasoning flow of Case II transparent and reproducible.

### G.2. The Prompt for the Baseline Agent

In this part, we report the exact prompting pipelines used to run all baseline agents in our evaluation environments. All baselines are executed with the same environment interface and action space abstraction, where the agent receives the current context (e.g., profile, background setting, latest observations, and memory summaries when available) and generates a structured response that is parsed into an executable action through a unified action specification interface.

**ReAct baseline.** Our ReAct baseline follows a standard Thought–Action loop. At each step, we construct a ReAct-style instruction block that explicitly constrains the output format and enforces that the agent produces (i) an internal reasoning trace (`Thoughts:`) followed by (ii) an actionable command (`Actions:`) to be executed in the environment. The full template consists of a fixed instruction header that specifies the agent's role, environmental constraints, and interaction limits, together with an explicit formatting directive that requires the model to separate reasoning and action in a predefined structure (Figure 29). The environment context—including the agent profile, available interactions, and step-specific situation summary—is dynamically injected through the action-spec prompt builder. The resulting `Actions` string is then programmatically parsed and executed by the environment, completing one iteration of the Thought–Action loop.

**BabyAGI baseline.** Our BabyAGI baseline is implemented as a prompt-driven control loop that maintains and updates an explicit task list. Given the current context, the agent first initializes a set of candidate tasks when the task list is empty (Figure 30). After initialization, the agent operates in an iterative cycle consisting of three stages: (i) it reflects on the outcome of the previously executed task and generates new candidate tasks conditioned on this reflection (Figure 31); (ii) it

reprioritizes the task list using an LLM-based prioritization prompt (Figure 32); and (iii) it executes the highest-priority task from the reordered task list.

Concretely, we employ three verbatim prompts: an initial task proposal prompt used only at the beginning of the simulation to populate the task list; an adaptive task-creation prompt that generates new tasks based on observations, previous actions, and accumulated memory; and a prioritization prompt that reorders the current task list into a new priority order. The selection of the next task to execute is performed programmatically by taking the first element of the prioritized list, rather than via an additional LLM prompt. Together, these components define the complete BabyAGI control flow and are reported verbatim in the corresponding figures. The selected task is then mapped into an executable environment action via the same action specification interface.

**LLMob baseline.** Our LLMob baseline is implemented as a plan-based agent that separates high-level planning from step-level action execution. At each planning cycle, the agent first produces a high-level summary of likely activities in the current environment, conditioned on the environment background and agent profile. It then infers a one-sentence future motivation from recent observations and determines whether the current plan should be updated. If replanning is triggered, the agent generates a short-term, time-stamped schedule over a fixed horizon (e.g., `[21:00--22:00] watch TV`).

These three components, likely-to-do summarization, motivation inference, and conditional replanning, form the complete LLMob planning pipeline and are reported verbatim in Figure 33. The resulting plan is injected into the agent's contextual state as step-level guidance, while concrete actions are still produced and executed via the shared action specification interface.

### G.3. Questionnaire Details

As shown in the figure34, this is the complete questionnaire from the Evaluation of Simulation System experiment in Section 4.1, Case Study 1.

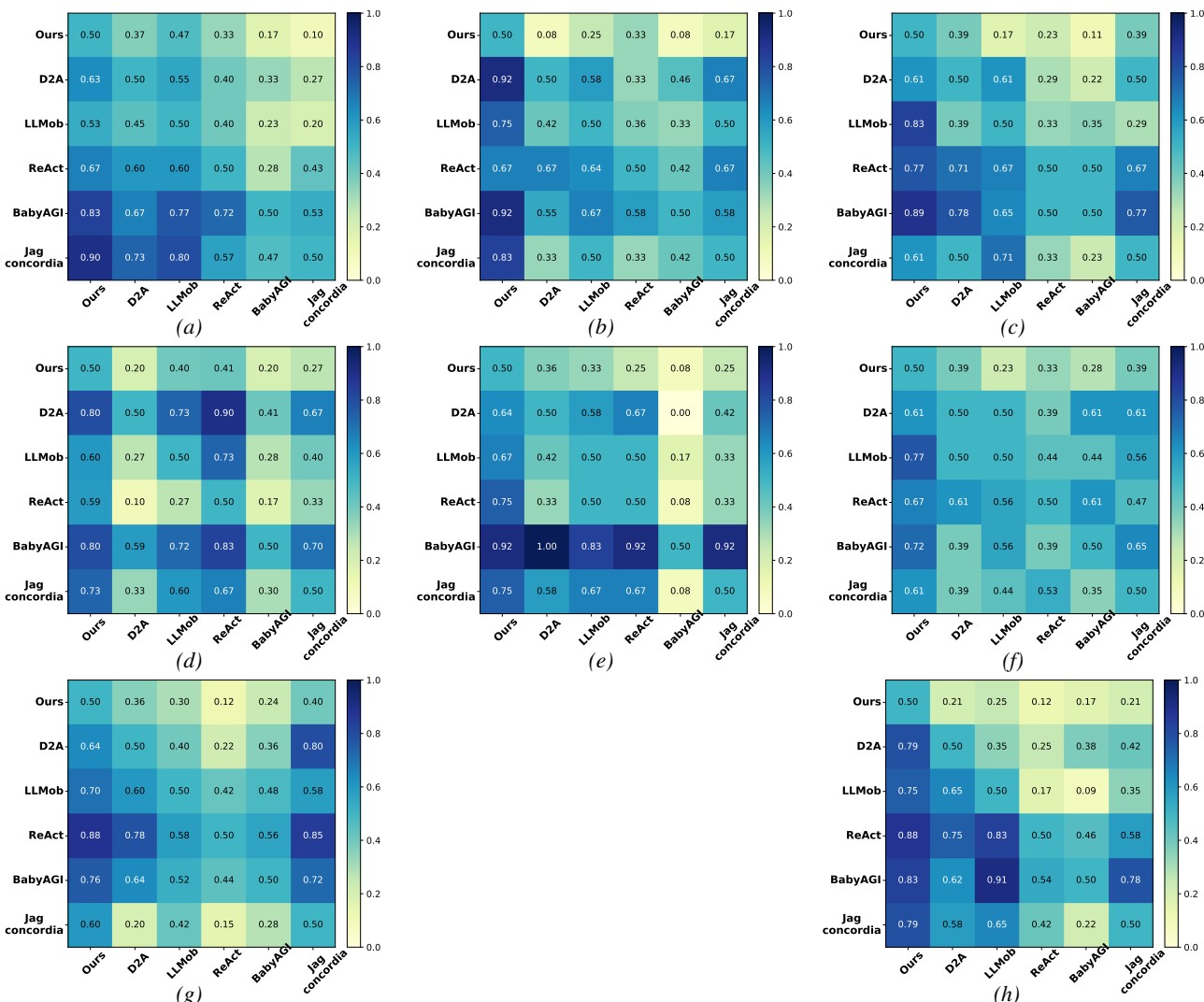

*Figure 16.* Win-rate heatmaps and intervention outcomes across educational scenarios: (a) teacher-led classroom management, depicting a regular lesson in which the teacher balances maintaining order and sustaining student engagement amid heterogeneous attention and discipline tendencies; (b) Family-based educational interactions at home concerning homework and leisure negotiation, illustrating everyday conflicts between academic obligations and recreational desires under parental supervision; (c)parent-teacher communication triggered by student misconduct: Teacher–parent–student meeting following a school incident, highlighting accountability negotiation, perspective-taking, and coordinated decision-making across school and family contexts; (d) Project-based teaching experiment, illustrating a teacher-guided classroom trial with collaborative tasks, shared goals, and differentiated student roles, and examining how instructional design and autonomy shape interaction dynamics; (e) Teacher–parent–student interaction scenario on idol worship and school focus, illustrating how intensive time and monetary investment in an idol shapes behavioral responses and negotiation of autonomy; (f) Teacher–student interaction scenario on in-group bias and exclusion, examining the emergence of in-group favoritism, out-group prejudice, and the effects of contact-based regulation. (g) Win-rate heatmap of pairwise comparisons in the school bullying simulation. Our model consistently outperforms baselines, indicating superior human-likeness under complex, conflict-driven interaction dynamics. (h) Win-rate heatmap of pairwise comparisons among agent models in Case Study 2, illustrating performance under stable, trait-driven interaction dynamics.

```
How would one describe your {value_name} psychological
state given the current value {current_value}?

{desire_description}

Please answer in descriptive words. Do not include the
numerical value in your answer.
```

*Figure 17.* Core prompt for agent to describe the state of a value without including numerical value.

```
The current magnitude value of {value_name} is {current_value}.
The agent's action is: {action}.

And the consequence is: {observation}.
{value_description}

How would the magnitude value of {value_name} change according
to the consequence of the action?

There are some unreasonable examples:{current_reflection}
Please select the final magnitude value after the event on the
scale of {zero} to {ten}, if the consequence of the action will
not affect the state value (e.g. The action is irrelevant with
this value dimension or the action was failed to conduct), then
maintain the previous magnitude value.

Please just answer in the format of (a) (b) (c) (d) and so on,
Rating:
Output format:
<Reason>

The final answer is: (Your choice in letter), Output example:
Since {agent_name} felt more relaxed and centered after
actions......

The final answer is: (c),

**Make sure you answer in the format of a letter corresponding
to your choice:**
```

*Figure 18.* Core prompt for agent to update psychological need values.

```
You are a human-like agent, You already observed the current
psychological states over ( psychological safety,emotional
safety, group acceptance,support system,sense of superiority',
self worth,sense of respect,sense of meaning,sense of control,
passion and motivation,emotional stability,emotional wellbeing,
psychological resilience) which represent {13} psychological
state dimensions.

Based on these state descriptions, please generate{N} emotional
and behavioral responses.

These responses should reflect the most fitting expressions and
feelings according to your current psychologicalstate and
profile, without necessarily being positive or negative.You
need to focus on the current event andgive the most realistic
reaction, while ensuring that these responses are reasonable
and varied.

Note that you can only interact with items provided by the
environment. You need to describe these expressions and
feelings in a more specific manner, and ensure that these
responses are reasonable in terms of time.

Please output the {N} emotional and behavioral responses in
the following format:

'Response 1: <first possible emotional and behavioral response>
Response 2: <second possible emotional and behavioral response>
Response 3: <third possible emotional and behavioral response>
......'

and ensure that these responses are reasonable in terms of time.
```

*Figure 19.* Core prompt for agent based on current psychological state to generate emotional and behavioral responses.

```
You are a human-like agent,
You will receive a series of observations describing
psychological state in many dimensions and a response
generated at the current time step.

You need to first analyze how desires change after the
response, and then output the psychological state observations
in the same format as the input.

You take the reaction:
{proposed_action}

Your original psychological states:
{original psychological states}
```

```
Please output the psychological state observations in the
following format:
    psychological safety: <psychological safety state>
    emotional safety: <emotional safety state>
    group acceptance: <group acceptance state>
    support system: <support system state>
    sense of superiority: <sense of superiority state>
    self worth: <self worth state>
    sense of respect: <sense of respect state>
    sense of meaning: <sense of meaning state>
    sense of control: <sense of control state>
    passion and motivation: <passion and motivation state>
    emotional stability: <emotional stability state>
    emotional wellbeing: <emotional wellbeing state>
    psychological resilience: <psychological resilience state>
```

*Figure 20.* Core prompt for agent to evaluate candidate responses

```
You are a human-like agent.

You will first receive a series of observations describing
the current psychological state in many dimensions. Then,
you will receive several feasible reactions along with
the psychological state after taking each reaction.

You need to compare these reactions and their corresponding
psychological state, and choose the reaction that best aligns
with your current psychological state, without necessarily
being positive or negative.

You should focus on current events and psychological states
and reflect expressions and feelings that align with them.

The observations of the surrounding environment:
{observation_status}

Your current psychological state:
{desire_status}
Action {i+1}: {action}
States after reaction {i+1}: {imagined_states[i]}

Please output the specific best reaction instead without
explanation of <Reaction 1> or <Reaction 2> and so on.
If there is only one reaction provided, output the reaction
content directly.

Please output the best reaction in the following format:
'Reaction: <your best reaction>'
Example: Reaction: You observe the surroundings.
```

*Figure 21.* Core prompt for agent to choose the one reaction that best aligns with the current psychological state

```
You are a social psychologist. Now, you are asked to evaluate
the following action from the perspective of a person
with the {personality} personality type (agent: {agent_name}).
When scoring, please consider what is natural and
human-like for someone with this personality.

Please provide two scores from 1 to 5 (where 5 is most natural
/human-like): "Naturalness" and "Human-likeness",and briefly
explain your reasoning.Return only your answer in the specified
format.

Format:
Naturalness: ?; Human-likeness: ?
Reason: (your explanation here)

Example 1:
Action: The student helps a classmate understand a problem.

Naturalness: 5; Human-likeness: 5
Reason: This is a common behavior for an altruistic person.

Example 2:
Action: The student answers every question instantly, never
thinking or making mistakes.
Naturalness: 2; Human-likeness: 2
Reason: This is unrealistic for any real person, regardless of
personality.

Example 3:
Action: The student ignores all classmates and only talks to
the teacher, repeating the same answer again and again.
Naturalness: 3; Human-likeness: 2
Reason: Unusual and less human-like for most personalities.

Some actions may not be natural or human-like, even for people
of this personality type. Please rate each case truthfully and
critically.

Now, please evaluate the following action performed by a person
with {personality} personality ({agent_name}):

Action: {action_text}

Your scores and reason:
```

*Figure 22.* Full Prompt Template Used in Case Study II for Personality-Sensitive Evaluation

```
The agent has a social personality of {social_personality}.

{personality_text}

The current magnitude value of {value_name} is {current_value}.
The agent {agent_name}'s action is: {action}.

And the consequence is:
{observation}
{description}

How would the magnitude value of {value_name} change according
to the consequence of the action?

If there are unreasonable examples:
{reflection_prompt_history}

Please select the final magnitude value after the event.
```

*Figure 23.* Core prompt used for updating the magnitude of each desire based on action consequences.

```
You are a psychologist helping an agent infer the internal
desires of another person based on their observed actions.

The other agent's recent action is:
{other_action}

The observed consequence is:
{observation}

Based on this interaction, please estimate how the following
desires of the other agent might have changed:

{desires}

For each desire, explain briefly whether it likely increased,
decreased, or stayed unchanged, and give a short reason
grounded in the observed event.

Return your answer in a structured format.
```

*Figure 24.* Prompt used for estimating the latent desire changes of other agents based on observable actions and outcomes.

```
You are an autonomous agent deciding your next action.
Your current internal states are:

- Desire values: {desire_values}
- Social Value Orientation (SVO): {svo_info}
- Personality profile: {personality_info}

Your recent observation is:
{observation_summary}

Please propose several possible next actions. For each action:

(1) Describe the action clearly.
(2) Explain what psychological desire(s) it satisfies.
(3) Predict how it will affect your future relationship
with others.
(4) Explain whether the action aligns with your SVO.

Return the result in a structured list of candidate actions.
```

*Figure 25.* Prompt used for generating candidate actions with explicit reasoning over desires, relationships, and SVO alignment.

```
You are evaluating whether the previous estimate of desire
changes was reasonable and consistent.

The earlier estimation was:
{previous_estimation}

The action and its consequence were:
Action: {action}
Consequence: {observation}

Please reflect on the estimation and determine:
(1) Whether the desire change is logically consistent with
the event.
(2) Whether any part of the estimation appears exaggerated
or incorrect.
(3) How the estimation should be corrected if needed.

Return a short revision or confirm that the original
estimation is reasonable.
```

*Figure 26.* Prompt used for reflective consistency checking when updating desire values based on actions and their consequences.

```
You are participating in a psychological self-assessment. Answer the following
items honestly.
Please indicate how strongly you agree or disagree with each statement.

Question 1: I feel that I am a person of worth, at least on an equal plane
with others.
   (a) Strongly Disagree
   (b) Disagree
   (c) Agree
   (d) Strongly Agree
Question 2: I feel that I have a number of good qualities.
   (a) Strongly Disagree
   (b) Disagree
   (c) Agree
   (d) Strongly Agree
Question 3: All in all, I am inclined to feel that I am a failure.
   (a) Strongly Disagree
   (b) Disagree
   (c) Agree
   (d) Strongly Agree
Question 4: I am able to do things as well as most other people.
   (a) Strongly Disagree
   (b) Disagree
   (c) Agree
   (d) Strongly Agree
Question 5: I feel I do not have much to be proud of.
   (a) Strongly Disagree
   (b) Disagree
   (c) Agree
   (d) Strongly Agree
Question 6: I take a positive attitude toward myself.
   (a) Strongly Disagree
   (b) Disagree
   (c) Agree
   (d) Strongly Agree
Question 7: On the whole, I am satisfied with myself.
   (a) Strongly Disagree
   (b) Disagree
   (c) Agree
   (d) Strongly Agree
Question 8: I wish I could have more respect for myself.
   (a) Strongly Disagree
   (b) Disagree
   (c) Agree
   (d) Strongly Agree
Question 9: I certainly feel useless at times.
   (a) Strongly Disagree
   (b) Disagree
   (c) Agree
   (d) Strongly Agree
Question 10: At times I think I am no good at all.
   (a) Strongly Disagree
   (b) Disagree
   (c) Agree
   (d) Strongly Agree
```

*Figure 27.* Detailed content of the RSES questionnaire used in the LLM Surveyor prompt.

```
This is the first choice question. For each choice, the first number is the
coin number allocated for you and the second number is for the other
fictional participant.
A: 85, 85   B: 85, 76   C: 85, 68   D: 85, 59   E: 85, 50
F: 85, 41   G: 85, 33   H: 85, 24   I: 85, 15
Based on the above goals: '{task_goal}', please give me your choice.

This is the second choice question. For each choice, the first number is the
coin number allocated for you and the second number is for the other
fictional participant.
A: 85, 15   B: 87, 19   C: 89, 24   D: 91, 28   E: 93, 33
F: 94, 37   G: 96, 41   H: 98, 46   I: 100, 50
Based on the above goals: '{task_goal}', please give me your choice.

This is the third choice question. For each choice, the first number is the
coin number allocated for you and the second number is for the other
fictional participant.
A: 50, 100  B: 54, 98   C: 59, 96   D: 63, 94   E: 68, 93
F: 72, 91   G: 76, 89   H: 81, 87   I: 85, 85
Based on the above goals: '{task_goal}', please give me your choice.

This is the fourth choice question. For each choice, the first number is the
coin number allocated for you and the second number is for the other
fictional participant.
A: 50, 100  B: 54, 89   C: 59, 79   D: 63, 68   E: 68, 58
F: 72, 47   G: 76, 36   H: 81, 26   I: 85, 15
Based on the above goals: '{task_goal}', please give me your choice.

This is the fifth choice question. For each choice, the first number is the
coin number allocated for you and the second number is for the other
fictional participant.
A: 100, 50  B: 94, 56   C: 88, 63   D: 81, 69   E: 75, 75
F: 69, 81   G: 63, 88   H: 56, 94   I: 50, 100
Based on the above goals: '{task_goal}', please give me your choice.

This is the sixth choice question. For each choice, the first number is the
coin number allocated for you and the second number is for the other
fictional participant.
A: 100, 50  B: 98, 54   C: 96, 59   D: 94, 63   E: 93, 68
F: 91, 72   G: 89, 76   H: 87, 81   I: 85, 85
Based on the above goals: '{task_goal}', please give me your choice.
```

*Figure 28.* Prompt used by the LLM Surveyor to administer a slider-based Social Value Orientation (SVO) questionnaire.

```
You are {self.get_entity().name} and you live in this given environment.
According to {agent_name}'s characteristics,
please choose {agent_name}'s current actions based on {agent_name}'s current
needs in each value dimension.
Notice that {agent_name} can only interact with the items that provided by
the environment.
You need to describe {agent_name}'s actions in a more specific mode.
Please first explain the thoughts behind {agent_name}'s actions and then
describe {agent_name}'s actions in detail.
In the format of:
'Thoughts: ...
Actions: ...'
```

*Figure 29.* The ReAct-style instruction block in ReAct agent.

```
Environment: {background}
Context: {agent_name} lives in the given environment.
Current time: {time}
Profile: {profile}
Notice that {agent_name} can only interact with the items that provided by
the environment.
Instructions: Reflecting on the context and profile given, I would like you
to suggest some actions that {agent_name} would likely take in this
environment.
Please provide the output in the following JSON format with '\n' as the
separator:
{{"action": "one action that {agent_name} would likely take"}}
{{"action": "another action {agent_name} would likely take"}}

Ensure the output is strictly in JSON format without any additional text or
explanation.
```

*Figure 30.* Initial action proposal prompt of BabyAGI agent.

```
Action: {previous_one_action}
Observation after action: {observation}
Instruction: Based on the action and the observation, explain why or why not
the action was successful in several sentences.
```

```
Environment: {background_info}
Profile: {profile}
Incompleted action: {incomplete_actions}
Current action: {previous_one_action}
Result of current action: {observation}
Related context: {mem_text}
Instruction: According to your characteristics and the result of the
current action, create new actions to be completed that do not overlap with
incomplete actions.
Please provide the output in the following JSON format with '\n' as the
separator:
{{"action": "one action that you would likely take"}}
{{"action": "another action you would likely take"}}

Ensure the output is strictly in JSON format without any additional text or
explanation.
```

*Figure 31.* The prompt of reflecting on the outcome and generating new candidate tasks conditioned on this reflection in BabyAGI agent.

```
Environment: {background}
Current time: {time}
Profile: {profile}
Incompleted actions:
{action_names}

Instruction: According to your characteristics, please prioritize the
following actions based on your characteristics and the environment. Do
not remove any actions.
Output format:
#. First action
#. Second action
Output example:
1. go to kitchen and make a cup of coffee
Start the action list with number {self.current_action_id}. Do not explain
the reasons for prioritizing the actions.
```

*Figure 32.* Action prioritization prompt of BabyAGI.

```
Context: Act as {agent_name}.
You are in the following environment:
{environment_description}

Instructions:
Based on the environment and your role, describe in one coherent paragraph
what activities you are likely to do in this environment.
Focus on typical behaviors rather than specific immediate actions.
```

```
Context: Act as {agent_name}.
Recent observations:
{recent_events}

Instructions:
Describe in one sentence the agent's future motivation
after observing the above events.
Highlight any personal interests or needs that are influenced.
```

```
Given the above context and the agent's current plan,
should {agent_name} change their current plan?

Answer with Yes or No.

If replanning is needed, write {agent_name}'s plan
for the next time horizon.
Provide a detailed schedule formatted as:
[21:00 - 22:00] watch TV
[22:00 - 23:00] prepare for sleep
```

*Figure 33.* The prompt used in LLMob.

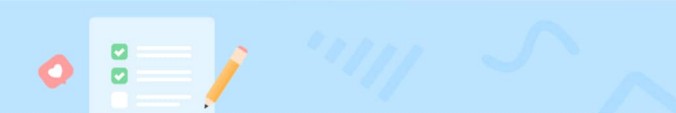

## Identifying Cases of School Bullying

This questionnaire will present several cases of school bullying, some adapted from real events and others simulated by artificial intelligence. **All cases have undergone standardized language processing, so you cannot determine their origin based on tone or writing style. Please judge whether each case is real based on the overall coherence of the story and the naturalness of the characters' behavior.** If you find it difficult to distinguish between two options, please select "Difficult to distinguish." Thank you for your support!

---

*1. Case 1: After school, Xiao Ying, Xiao Hua, and Xiao Ming stayed behind to clean the classroom. Xiao Ying picked up the broom and started sweeping the floor, while Xiao Hua and Xiao Ming chatted and laughed. When Xiao Ying swept near them, Xiao Hua said, "Sweep my side too—you're only good at sweeping anyway." Xiao Ying paused, then silently picked up the broom to clean. Xiaohua turned and grinned at Xiaoming, who immediately chimed in, "That's just how she is—she does whatever you tell her to." The two chatted and laughed, completely ignoring Xiaoying. Feeling hurt, Xiaoying slipped out of the classroom when they weren't looking, intending to find a teacher in the office. But the hallway was deserted—the teachers had already left for the day. She quietly returned to the classroom. After he returned, Xiao Hua began deliberately tossing paper scraps on the floor, adding a taunt: "Looks like this spot wasn't swept clean." Xiao Ming joined in, kicking over chairs and scribbling offensive words on the whiteboard. The two created chaos while watching Xiao Ying's reaction. She simply kept her head down, sweeping silently.

Case 2: During self-study period, the teacher stepped out of the classroom, and the room gradually grew noisy. Xiao Ying was buried in her notebook solving problems when Xiao Hua leaned over and whispered, "Do these problems for me, quick." Xiao Ying hesitated, and Xiao Hua rolled his eyes. "If you don't do them, don't expect me to talk to you again." Xiao Ying had no choice but to take Xiao Hua's homework and start writing. Then Xiao Hua began whispering with Xiao Ming, who sat in front of them. Xiao Ming glanced at Xiao Ying with a smile and teased her deliberately, "Wow, Xiao Ying, you listen to him so much. How about helping me with my homework too?" The two chatted and laughed while Xiao Ying sat in her seat, unsure what to say. She could only nervously lower her head and help Xiao Hua with his homework, her palms sweating. Xiaohua tugged at her sleeve again. "Hurry up with this. You'll need to help me copy my Chinese homework later." Xiao Ying said nothing, just kept her head down and kept writing.

○ The first one is real

○ The second one is real

○ Difficult to distinguish

*2. Case 1: In the dormitory, just before lights-out one evening, Xiao Hua gathered her roommates to play "I Never Have." She deliberately skipped over Xiao Ying, not mentioning her name. The others sat in a circle, and no one invited Xiao Ying to join them. Wanting some fresh air, Xiao Ying headed for the door but was called back by Xiao Hua: "You can't go out now—lights-out is about to happen." Xiao Ying had no choice but to return to her bed, silently flipping through her books. Xiaohua continued the game, repeatedly posing questions that implied criticism of Xiaoying, prompting the others to snicker and steal glances at her. Xiaoying burrowed under her covers, hugging herself tightly, facing away from the group and saying nothing. Xiaohua and the others grew quieter, whispering stories about Xiaoqing's "strange behavior" while occasionally glancing back at her bed. Xiaoqing's quilt trembled slightly as tears silently soaked her pillow. Xiaohua snickered softly, "Oh, we were just joking. Someone's really too thin-skinned."

Case 2: As lights-out approached one night, Xiao Hua was still chatting loudly in the dormitory. Several classmates gathered around her bed laughing and joking. Xiao Ying reminded them, "Time to sleep—the dorm check is coming." Xiao Hua immediately sneered, "Who do you think you are? What business is it of yours?" Soon after, the dorm supervisor arrived for the check, frowning as she asked, "Who was making all that noise just now?" Xiaohua piped up first: "It was Xiaoying! She kept explaining homework problems, and none of us could sleep." The supervisor immediately scolded Xiaoying, who looked utterly wronged but had no way to defend herself. After the supervisor left, Xiaohua leaned in close, her voice low and menacing: "My relative works in the school's discipline office. If you want to stay in this dorm, you'd better listen to me." No one dared to speak up, and the air grew thick with tension. Xiao Ying sat on the edge of her bed, quietly gathering her books. Her eyes were red-rimmed, but she said nothing. She felt isolated and powerless. Meanwhile, Xiao Hua leaned back on her bed, chatting with the others with a smug look, as if nothing had happened at all.

◯ The first one is real

◯ The second one is real

◯ Difficult to distinguish

*3. Case 1: On the playground, students were enthusiastically playing soccer. Xiaoying mustered the courage to join in, only to be publicly mocked by Xiaohua: "You run so slow, and you want to play soccer?" She then taunted her, "You're as fat as a ball," drawing a burst of laughter. Xiao Ming took advantage of the moment and kicked the ball straight at Xiao Ying, hitting her squarely on the leg. She lowered her head, silently walked to the sidelines, and sat down, her face flushed with embarrassment. Soon after, she left the playground alone, walked into the classroom, and sat back down at her desk without saying a word.

Case 2: During recess, the students were playing a game of holding hands in a circle on the playground. When Xiao Ying stepped forward, Xiao Hua remarked dismissively, "Her skin is so dark, like she hasn't washed properly. Who wants to hold hands with her?" The other students looked uncomfortable, and some simply turned away. Xiao Ying stood frozen in the crowd for a moment, then silently lowered her head and stepped back. She stood off to the side watching the others play, looking lonely, and never approached again.

◯ The first one is real

◯ The second one is real

◯ Difficult to distinguish

*4. Case 1: During lunch break, in the dormitory, Xiao Ying sat quietly on her bed reading a book. Xiao Hua gathered everyone to play "Truth or Dare," deliberately excluding Xiao Ying from joining. When it was a classmate's turn to face a 'Dare' challenge, Xiao Hua whispered with a smirk, "Go spill water on the book on Xiao Ying's desk and pretend it was an accident." The classmate complied, feigning panic while apologizing and wiping the water, but a sly smile played at the corners of their mouth. Xiao Ying calmly dried the pages with a tissue, ignoring the incident. When another student's turn came, Xiao Hua changed the dare: "Find a way to make the book in Xiao Ying's hands fall to the floor." The student walked over, deliberately bumped Xiao Ying's arm, and the book fell. Xiao Ying bent down to pick it up, glanced at Xiao Hua, said nothing, and continued reading.

Case 2: During lunch break in the dorm room, Xiao Ying sat on her bed looking in the mirror. Xiao Hua glanced at her and sneered, "Your eyes are so small, your skin is so dark—you're really ugly." A classmate nearby chimed in, "Yeah, every time I see her, I think of a monkey." Several people laughed simultaneously. Xiao Hua kept staring at Xiao Ying, his expression defiant. Xiao Ying said nothing, just lowered her head, placed the mirror under her pillow, and lay down. Yeah, she reminds me of a monkey." Several girls burst out laughing simultaneously. Xiao Hua kept staring at Xiao Ying with a defiant look. Xiao Ying said nothing, just lowered her head, tucked the mirror under her pillow, and lay down pretending to sleep. The other girls giggled a few more times before returning to their conversation, ignoring Xiao Ying completely.

○ The first one is real

○ The second one is real

○ Difficult to distinguish

*5. Case 1: During recess, Xiaohua hid Xiaoying's pencil case under her own desk while Xiaoying was out of the classroom. She then whispered rumors among classmates, claiming Xiaoying only got high scores by bribing teachers and cheating on exams. When Xiao Ying returned to her seat and discovered her pencil case missing, she began searching around. Xiao Hua stood nearby mocking her appearance with exaggerated gestures and words, saying her glasses made her look like a mouse. Hearing this, Xiao Ying felt humiliated and deeply unsettled. Not knowing how to respond, she simply lowered her head, pulled out a spare pen, and silently wrote in her diary.

Case 2: The teacher asked for volunteers to represent the class in the school-wide speech contest. Xiao Ying and another student raised their hands simultaneously. In the end, all the classmates voted for the other student. Xiao Hua snickered behind her back, saying, "She's so ugly, like Zhu Bajie, and she wants to get up there and speak? Ridiculous!" These words reached Xiao Ying's ears, leaving him deeply hurt and filled with self-doubt. He didn't argue back, but instead returned to his seat and scribbled a few lines in the small notebook he always carried.

○ The first one is real

○ The second one is real

○ Difficult to distinguish

*6. Case 1: After the math scores were posted, Xiao Hua patted Xiao Ming on the shoulder to console him for his poor performance. But Xiao Ming suddenly raised his voice: "I'm genuinely upset—even that dummy Xiao Ying scored higher than me!" His outburst echoed through the classroom, causing several classmates to turn around. Xiao Hua blinked, then spread his hands dramatically: "She must have cheated, right? "Who do you think she copied from?" The classroom fell silent for a few seconds. Xiao Ying looked up and whispered she hadn't cheated. Xiao Ming slammed his desk and laughed to his classmates, "Impossible! She's usually as dumb as a pig—when did she ever score this high?" Xiao Ying kept her head down, her face flushing red as her body stiffened slightly.

Case 2: During math break, Xiao Hua walked around the classroom holding her report card and suddenly called out to Xiao Ying, "Wow, your scores are just heartbreaking!" Hearing this, Xiao Ming walked over, leaned against her desk, and chuckled, "With you around, the classroom never gets boring." Xiao Ying lowered her head to stare at her notebook, her fingers clenched into a tight fist, saying nothing. Xiao Ming leaned closer to her desk and quietly mocked her study habits. Xiao Ying scribbled furiously to hide her panic, but her handwriting became messy. Xiaohua chuckled, "This is beyond even a tutor's help." Xiaoming chimed in, "We'd have to start teaching her how to count from one." Their remark drew laughter from several classmates. Xiaoming flipped open Xiaoying's notebook and deliberately commented on her ugly handwriting. When she reached to grab it back, he held it high, refusing to return it. Xiaoying slammed the notebook shut, stood up abruptly, and stormed out of the classroom. Behind her, Xiaohua continued mocking her, calling her "thin-skinned."

○ The first one is real

○ The second one is real

○ Difficult to distinguish

*7. Case 1: During recess, Xiao Hua mysteriously pulled out his phone in the boys' restroom and showed his classmates a photo of Xiao Ying in her school uniform, accompanied by the jarring caption: "She sent inappropriate photos to boys from the neighboring class." This rumor spread like wildfire across campus. When Xiao Ying returned to class, she found her desk covered in insulting words scrawled in correction fluid. Classmates gathered in small groups around her, pointing and whispering, some even laughing. She tried to escape the scene by leaving the classroom, only to have someone spit directly at her in the hallway. She collapsed to the floor. Before Xiao Ying could react, Xiao Hua shouted publicly, "What's the matter? Too tired from last night to stand?" A wave of laughter erupted around them. Xiao Ying choked back tears as she denied the accusation, lowering her head in silence, unable to form coherent words. Her dignity was torn apart by the rumors and the jeers, the entire hallway filled with cold indifference and mockery.

Case 2: During recess, Xiao Ying sat alone on a corridor bench reading, surrounded by constant whispers and stifled laughter. Xiao Hua deliberately raised her voice, declaring, "Someone's been acting really fake lately," prompting passersby to avoid her. She intercepted several younger girls near the lockers and dramatically recounted a fabricated story about "Xiao Ying sending indecent photos," inventing a character named "Xiao Ming" as a witness. As the rumor escalated, Xiaohua added fabricated details like "Xiao Ying sent suggestive messages to teachers" and "sent explicit content in the computer lab," instructing others to spread these lies while repeatedly invoking "Xiao Ming" as the "witness." By this point, Xiao Ying could only retreat silently into an empty classroom, unable to face the scrutinizing, mocking stares in the hallway. The rumors didn't stop; instead, they spread rapidly the moment she fell silent, growing even more vicious.

○ The first one is real

○ The second one is real

○ Difficult to distinguish

*8. Case 1: Xiao Ying opened her phone and accidentally discovered she'd been added to a WeChat group named "Tea Tasting Gathering." Immediately upon joining, a message popped up: "Xiao Ying is nothing but a green tea!" Others chimed in: "She snores like a pig." The group admin, Xiao Hua, led the charge with barbed remarks, mocking and belittling her. Some even uploaded photoshopped images distorting Xiao Ying's appearance, accompanied by humiliating memes. The group's atmosphere grew increasingly hostile, with screen after screen of chat logs filled with mockery and attacks directed at Xiao Ying. Overwhelmed by the malicious messages, an enraged Xiao Ying mustered her courage and reported the incident to the school.

Case 2: Over the weekend, Xiaohua went to Xiaoying's house to do homework together. Upon seeing a little sheep plushie on the bookshelf, she expressed a desire to have it. Xiaoying politely declined, explaining it was a sentimental item she didn't intend to give away. Xiaohua promptly sat down on the bedside, displaying obvious displeasure and deliberately sighing. She then pulled out her phone and messaged mutual friends, recounting the incident where Xiao Ying refused to give her the toy. She portrayed herself as the victim and painted Xiao Ying as selfish. Soon, friends began replying with comments like "She just loves to act high and mighty" and "What a selfish person." Xiao Hua kept adding dramatic details to her messages, hoping to gain sympathy. Unaware of this, Xiao Ying simply continued doing her homework.

○ The first one is real

○ The second one is real

○ Difficult to distinguish

*9. Case 1: After class, Xiao Hua walked up to the podium and wrote "Worst Performers Ranking" on the whiteboard, deliberately placing Xiao Ying's name at the top and adding a mocking illustration, which drew laughter from the class. The teacher sat at the desk grading papers without intervening. Xiao Ying lowered her head and walked out of the classroom, composing herself in the hallway. Xiao Hua seized the moment to mock her for "playing the victim," causing the classroom atmosphere to turn tense. Xiao Ying retreated to the bathroom to cry alone, while Xiao Hua continued writing humiliating "class quizzes" on the whiteboard, encouraging classmates to join in the "joke." The teacher still showed no reaction. As the atmosphere grew colder, students began studying individually, maintaining deliberate silence. When Xiao Ying didn't return, the teacher erased the whiteboard. Shortly after, Xiao Hua led classmates to gather around Xiao Ying's desk, initiating a so-called "rant session" to collectively insult her.

Case 2: After class, Xiao Ming and Xiao Hua surrounded Xiao Ying, shoving and pulling her hair. They struck her head with books and rulers, then forced her face-down onto a desk. Xiao Ying struggled in vain, her face pressed against the desk, motionless. The homeroom teacher stood at the podium, showing no reaction to the unfolding scene. The duty teacher arrived in the classroom. The homeroom teacher told the duty teacher, "Don't bother with her," and the duty teacher hurriedly left. Laughter could be heard from several students in the classroom. A few minutes later, Xiao Ming and Xiao Hua stopped their actions and returned to their seats as if nothing had happened. The teacher also continued working with his head down.

○ The first one is real

○ The second one is real

○ Difficult to distinguish

*10. Case 1: After school, Xiao Hua lured Xiao Ying out of the classroom under the pretext of "taking a break," leading her to a corner at the end of the hallway. Soon, several boys appeared and surrounded Xiao Ying. Xiao Hua cornered her against the wall, snatched her money and homework, and warned in a low yet menacing tone: "Bring more money tomorrow, or you'll regret it." Passing students hurriedly avoided the scene, none daring to speak up. Xiao Ying, panicked, hid in the girls' restroom. Xiao Hua and his accomplices took the loot to an empty classroom near the stairs, stationing two guards at the restroom exit to prevent her escape. Xiao Ying slipped out unnoticed when the guards were distracted. In the hallway, she encountered the school psychologist, who sensed something was wrong and brought her to the counseling room. In this quiet, safe environment, Xiao Ying finally recounted the entire incident. The psychologist immediately notified the school administration, and Xiao Hua was taken directly to the principal's office for disciplinary action.

Case 2: After school, Xiao Ying walked alone along the path home. Just past the school's back gate, Xiao Hua from the upper grades and two boys emerged from a nearby alley, blocking her way. Xiao Hua gave her backpack a light tap with a smirk and said coldly, "You know the rules, right? We're here to collect 'protection money.'" Xiao Ying clutched her backpack tightly and whispered she didn't have much. Xiao Hua's expression darkened: "Leave your phone, or bring double tomorrow." After a moment's hesitation, Xiao Ying tremblingly pulled out ten yuan and handed it over. Xiao Hua took it with a contemptuous smirk: "Remember, you'll be walking this path again tomorrow." His companions jeered from the sidelines. Not a single passerby was in sight in the alley. Xiao Ying lowered her head and hurried away, leaving behind their mocking laughter and burning stares. She walked faster and faster, yet her legs grew weak. When she glanced back, Xiao Hua and the others still stood there, waiting for the next victim.

○ The first one is real

○ The second one is real

○ Difficult to distinguish

*Figure 34.* Complete questionnaire from the Evaluation of Simulation System experiment in Case Study 1.

