# OpenReview forum: "EduMirror: Modeling Educational Social Dynamics with Value-driven Multi-agent Simulation"
_ICML.cc/2026/Conference — ICML 2026 regular_

### Official Review · Reviewer_oFjf · 2026-02-20

**Soundness:** 3
**Presentation:** 3
**Significance:** 3
**Originality:** 3
**Overall Recommendation:** 4
**Confidence:** 3

**Summary:**

This paper introduces EduMirror, a comprehensive multi-agent simulation framework designed to “mirror” and analyze the causal mechanisms of educational social dynamics. As direct utilization of LLMs as social agents suffers from inconsistency between psychological states and behavior, this paper aims to to enable realistic, interpretable, and measurable social simulation for educational study. This approach consists of , which supports a curated library of 20 pre-designed educational scenarios, and a value system to quantify the state of   each agent for adaptive behaviour generation. Case studies demonstrate EduMirror’s capacity to reproduce realistic, coherent narratives.

**Compliance With Llm Reviewing Policy:**

Affirmed.

**Final Justification:**

This paper addresses a critical and highly significant issue in the social sciences. The rebuttal has fully resolved my questions and I lean toward acceptance.

**Key Questions For Authors:**

1. Is there any behaviors or phenomenon that is unexpected or unseen in the real world?

**Limitations:**

Yes. Limitation is discussed in this paper.

**Strengths And Weaknesses:**

Strength:
1. This paper addresses a critical and highly significant issue in the social sciences: educational and social problems such as school bullying or peer pressure experienced by teenagers during their primary and secondary education. By designing and implementing EduMirror, it demonstrates the potential of large-scale model technology to tackle such challenges where real-world experiments are difficult to conduct.
2. The case studies are thorough and intuitively presented. The simulations of school bullying and peer group cooperation are detailed, logically structured, and clearly connected to established social science theories.

Weakness:
1. Limited Technical Novelty. While the system is well-engineered and application-driven, the technical components largely build upon existing multi-agent paradigms. The paper lacks methodological innovation at the algorithmic or modeling level.
2. Reliance on LLM-Based Evaluation and Potential Validity Concerns. The dual-track measurement protocol depends heavily on LLM-based assessors to transform qualitative interactions into quantitative metrics. However, the reliability, bias, and reproducibility of such LLM-as-judge evaluations are not rigorously validated.

---

> ### Author Rebuttal · Authors · 2026-03-31
>
> > ***W1 (Limited technical novelty)** **:** Lack of methodological innovation at the algorithmic or modeling level.*
> >
>
> Thank you for recognizing the engineering quality of our framework. We respectfully argue that our novelty lies in three mechanism-level contributions to educational simulation: cognition-inspired agent architecture, systematic intervention analysis, and their integration into a cohesive framework.
>
> **(1) Problem significance and educational value.** Educational intervention design traditionally relies on small-scale human studies or simplistic computational models that cannot capture complex social dynamics. Existing multi-agent frameworks lack the psychological realism required for authentic educational scenarios. EduMirror fills this gap by enabling large-scale, repeatable simulation with cognition-inspired agents, providing a testbed for intervention design that delivers actionable insights for educational practice.
>
> **(2) Cognition-inspired agent architecture.** Unlike general-purpose LLM agents relying on implicit prompting, we introduce a value-driven architecture specifically designed for educational social simulation. The framework explicitly formalizes Social Value Orientation and psychological needs (autonomy, competence, relatedness) as structured decision factors, grounding agent behavior in quantifiable internal states rather than ad-hoc prompting. This enables authentic social behaviors, including cooperation, competition, social adaptation, and emotionally realistic responses, essential for educational scenarios.
>
> **(3) Systematic intervention analysis.** Complementing the agent architecture, our dual-track measurement protocol jointly evaluates observable behaviors and latent psychological states, enabling causal analysis of intervention mechanisms that is virtually impossible in real-world settings due to ethical constraints. For example, we can isolate whether a bullying intervention succeeds through emotional support versus social status restoration. Such fine-grained, controlled analysis of educational dynamics is absent in existing multi-agent frameworks.
>
> > ***W2 (Reliability of LLM-based evaluation):**  The reliability, bias, and reproducibility of LLM-as-a-judge are not rigorously validated.*
> >
>
> This is a valid concern. We evaluate reliability via a stability analysis over 50 repeated runs.
>
> A `Mann–Whitney U test` yields **p = 0.0296 (< 0.05)**, indicating that performance differences are statistically significant. We further calculate the `Intraclass Correlation  Coefficient` (**ICC = 0.753**), indicating strong consistency and stable evaluation across multiple runs.
>
> To validate alignment with human judgment, we compared LLM rankings with human consensus over five models, based on evaluations from 20 human raters and five LLM judges. We report Kendall’s tau-b for human–LLM agreement and mean pairwise tau-b for inter-LLM consistency.
>
> We also assessed intra-judge repeatability by re-running each LLM judge 10 times on the same five models and computing ICC(3,1) and ICC(3,k) from model-level scores. Here, ICC(3,1) measures single-run reliability, while ICC(3,k) measures the reliability of the average across repeated runs.
>
> | Judge | **Ranking** | **tau-b vs Human** | **Mean tau-b vs LLMs** | ICC(3,1) | **ICC(3,k)** |
> | --- | --- | --- | --- | --- | --- |
> | Human | EduMirror > BabyAGI ≈ LLMob > D2A > React | 1.000 | - | - | - |
> | DeepSeek | EduMirror > LLMob > BabyAGI > D2A > React | 0.949 | 0.800 | 0.966 | 0.997 |
> | Qwen | EduMirror > BabyAGI > LLMob > React > D2A | 0.738 | 0.700 | 0.989 | 0.999 |
> | Claude | EduMirror > BabyAGI > LLMob > D2A > React | 0.949 | 0.750 | 0.995 | 0.999 |
> | GPT | EduMirror > LLMob > BabyAGI > React > D2A | 0.738 | 0.750 | 0.974 | 0.997 |
> | gemini | EduMirror > LLMob > BabyAGI > D2A > React | 0.949 | 0.800 | 0.975 | 0.997 |
>
> Overall, these results indicate that model-level realism rankings are broadly consistent across human raters and multiple LLM judges, and that the resulting rankings are highly stable across repeated evaluations.
>
> > ***Q1 (Unexpected behaviors):** Are there any unexpected or non-real-world behaviors observed in the simulations?*
> >
>
> We observed two categories of unexpected behaviors: (1) Overly rational emotional regulation: Victimized agents sometimes recovered unrealistically fast due to LLM safety alignment, exhibiting mature coping (e.g., mindfulness). We mitigated this through model selection and by grounding behavior generation in the agent’s psychological state, translating quantitative signals into contextualized descriptions to induce more realistic distress and delayed recovery. (2) Emergent competition: In the class monitor election, agents developed rivalrous strategies without explicit rules, consistent with sociological theories of emergent competition [1].
>
> **References:**
>
> [1] Deutsch, M. (1949). *A theory of cooperation and competition*. *Human Relations, 2*(2), 129–152.

---

> > ### Author Rebuttal · Reviewer_oFjf · 2026-04-01
> >
> > Thank you for the detailed rebuttal. I will maintain my positive score.

---

### Official Review · Reviewer_oswg · 2026-03-13

**Soundness:** 3
**Presentation:** 3
**Significance:** 2
**Originality:** 3
**Overall Recommendation:** 4
**Confidence:** 3

**Summary:**

The paper introduces EduMirror, a multi-agent simulator that studies the mechanisms of educational social dynamics. It integrates LLMs with a value-driven cognitive architecture and is applied to scenarios including school bullying and group cooperation. Results show that EduMirror generates scenarios aligned with established theories.

**Compliance With Llm Reviewing Policy:**

Affirmed.

**Final Justification:**

I would like to thank the reviewers for the detailed rebuttal. I will keep my current score.

**Key Questions For Authors:**

* Authors adapt a dual-track measurement protocol which relies on LLM rater and surveryor to evaluate behavior generated by LLMs. I wonder if this will bias the evaluation towards certain LLM-typical speech patterns
* I wonder how the framework handles the long-context issue in LLMs, given the amount of information to be provided to LLMs as memory. How will the model choose what to remember and what to forget?

**Limitations:**

yes

**Strengths And Weaknesses:**

Strengths:
* The framework is grounded on established social and psychological theories. The design is well motivated and also clearly described in the paper
* The paper integrates an intervention engine and expands scenarios to enable the what-if experimentation. This supports causal analysis and makes the framrwork more solid.

Weakness:
* The paper is a little dense to read, with too much information in the appendix instead of the main text. I understand that this is partially due to the page limit of the conference, but authors may consider including details like the dual-track measurement in the main text

---

> ### Author Rebuttal · Authors · 2026-03-31
>
> > ***W1 (Placement of crucial information):** Dense structure with crucial details (e.g., dual-track measurement) placed in the appendix.*
> >
>
> We thank the reviewer for this suggestion. In the revised version, we will move key components of the dual-track measurement protocol into the main text, while keeping implementation details in the appendix.
>
> > ***Q1 (Bias in LLM-as-a-judge):** Does the dual-track measurement bias the evaluation towards LLM-typical speech patterns?*
> >
>
> This is a very valid concern. To verify whether our protocol biases toward specific LLM stylistic patterns, we conducted an additional controlled experiment. Specifically, we generated agent behaviors using multiple LLMs (**DeepSeek-V3.1, Gemini-2.5-flash, GPT-5.4, Qwen-3.5-122b, and Claude-Sonnet-4.6**), and applied a unified style normalization process to reduce stylistic differences across models. We then compared these normalized outputs against baseline models using the same evaluation pipeline, and measured their win rates under pairwise comparisons. For each model pair, we conducted **20 independent pairwise comparisons**. We measure win rates based on pairwise comparisons, defined as: $\text{WinRate}(M, B) = \frac{N_{M > B}}{N_{\text{total}}}$, where $M$ denotes the evaluated model and $B$ denotes the baseline model. As summarized in the table below, the win rates remain highly consistent regardless of the underlying generation model:
>
> |  Baseline  |  DeepSeek (%)  |  Gemini (%)  |  GPT (%)  |  Qwen (%)  | Claude(%)  |
> | --- | --- | --- | --- | --- | --- |
> |  BabyAGI   |  70 |  80 |  75 |  70 | 80 |
> |  D2A       |  90 |  85 |  85 |  90 | 90 |
> |  LLMob     |  80 |  70 |  85 |  70 | 80 |
> |  ReAct     |  80   |  80 |  85 |  80 | 85 |
>
> The consistency of results suggests that our evaluation is robust to stylistic variations and primarily captures behavioral differences rather than surface-level language patterns.
>
> > ***Q2 (Long-context memory mechanisms):** How does the framework handle the long-context issue and select what to remember/forget?*
> >
>
> In our framework, long-context pressure is handled primarily by **externalizing memory** and injecting only a small working set into the prompt, rather than by passing the full interaction history to the LLM. **To manage long context**, the architecture stores observations and formative memories in an associative memory bank, while the prompt itself is constructed from a bounded observation window and a small set of retrieved memories. In the default agent used in this project, the direct event history is capped (history_length = 100), recent-memory reasoning typically uses only the most recent 12 items, and the semantic memory module first asks the LLM to summarize the current state and then retrieves only the top relevant memories (e.g., top 10) based on embedding similarity. In the value-agent branch, the same idea is preserved through short observation summaries over a limited set of recent memories. Thus, the framework reduces context length through a combination of **windowing, summarization, and retrieval-based selection**.
>
> **Regarding memory selection, what the model “remembers” is not the full memory store, but the dynamically selected subset used for the current decision step.** Specifically, **what to remember** is driven mainly by recency and semantic relevance: recent events are surfaced through bounded recent-memory queries, while older information is recalled only if it is semantically similar to the current summarized situation. Conversely, regarding **what to forget**, the current implementation does not perform strong deletion-based forgetting; the memory bank largely accumulates entries and removes only exact duplicates. Instead, “forgetting” occurs functionally at the prompt-construction stage: information that is neither recent nor semantically relevant is simply not re-inserted into the model’s active context, effectively rendering it “forgotten” for the current interaction.

---

> > ### Author Rebuttal · Reviewer_oswg · 2026-04-04
> >
> > I would like to thank the reviewers for the detailed rebuttal. I will keep my current score.

---

### Official Review · Reviewer_Nqz4 · 2026-03-13

**Soundness:** 3
**Presentation:** 2
**Significance:** 3
**Originality:** 2
**Overall Recommendation:** 4
**Confidence:** 4

**Summary:**

This paper introduces EduMirror, a multi-agent simulation platform designed to model educational social dynamics using large language model–driven agents grounded in psychological and social value frameworks. The system integrates a value-driven cognitive architecture with a scenario-based simulation environment to enable the study of complex educational phenomena such as bullying, cooperation, and classroom social interactions. Overall, this research's principal topic pertains to the use of LLM-driven multi-agent simulations as a tool for studying educational social dynamics and enabling in silico experimentation in contexts where real-world experimentation may be ethically difficult.

**Compliance With Llm Reviewing Policy:**

Affirmed.

**Key Questions For Authors:**

1. The experiments focus on relatively small-scale simulations with limited numbers of agents. The authors should discuss whether the framework can scale to larger social systems such as entire classrooms or schools.

2. The paper claims that EduMirror enables causal analysis through counterfactual simulation. The authors should clarify how causal conclusions are validated and whether the simulation outcomes correspond to empirical findings from educational research.

**Limitations:**

See weakness.

**Strengths And Weaknesses:**

Strength:

1. The paper addresses an important and underexplored research problem by attempting to model educational social dynamics using LLM-based multi-agent systems. The motivation is well articulated, particularly the argument that traditional observational and experimental methods are limited when studying sensitive educational phenomena such as bullying or peer influence.

2. The experimental section includes multiple case studies that demonstrate the flexibility of the platform. For example, the bullying simulation analyzes how different intervention strategies affect the victim’s psychological needs over time, while the social interaction experiments explore cooperation–competition dynamics in classroom settings. These examples illustrate potential applications of the framework in educational research.

Weakness:

1. The evaluation metrics are also somewhat subjective. Measures such as naturalness, plausibility, and behavioral realism are assessed through automated judgments rather than objective quantitative metrics or controlled behavioral experiments.

2. The simulations involve relatively small groups of agents and simplified educational scenarios. Although the paper claims to model educational social dynamics, the scale and complexity of the simulations remain limited compared with real educational environments.

---

> ### Author Rebuttal · Authors · 2026-03-31
>
> > ***W1 (Subjective evaluation metrics):** Concerns about reliance on automated over human or objective evaluation.*
> >
>
> Thank you for raising this concern. While metrics such as naturalness and plausibility are inherently qualitative, our evaluation is not based solely on automated judgments.
>
> To assess behavioral realism, we conducted a human evaluation (Section 4.3) with 152 participants in a Turing-test-style task (`Table 8`). While *LLM-as-a-Judge* is validated in prior work [1,2,3], we further verify it in our setting by comparing LLM outputs with human judgments (Appendix F.4), showing strong alignment (up to **100% agreement** depending on consensus level).
>
> To further substantiate this finding, we replicated the same human evaluation protocol in `Study 2` with 21 participants. Each participant evaluated 20 randomly sampled cases, and results were aggregated at the case level based on inter-rater agreement. Most cases fell into the high-consensus category (**89.2%**), with the remainder in the moderate range and none in the low-consensus range, indicating robust behavioral realism.
>
> | Consensus Category | # of Cases | Agreement (%) |
> | --- | --- | --- |
> | High (>75%) | 14/20 | 89.2 |
> | Mod.  (50–75%) | 6/20 | 57.7 |
> | Low (<50%) | 0 | — |
>
> We further conducted an agreement analysis between human raters and LLM judges on both our model and baselines. Results show consistently high agreement across human–LLM and inter-LLM evaluations (see **Response to oFjf (W2)**for more details).
>
> > ***W2 & Q1 (Scalability to larger groups):** Can the framework scale to larger, complex systems?*
> >
>
> The framework is not limited to short or simplified interactions. EduMirror adopts a modular scenario design with reusable environments and agent profiles, enabling scalable simulations by adding agents, locations, and contextual elements without modifying the core architecture.
>
> We conduct a preschool simulation with a teacher and students across interconnected settings (e.g., gate, classroom, playground, corridor, nap room) under a structured daily schedule. We evaluate `Naturalness, Coherence, and Plausibility`, and introduce `Developmental Typicality`, grounded in Piagetian and Kohlberg’s theories [4,5], to assess age-appropriate reasoning. We run simulations with **5, 15, and 30 agents**, with results shown below; full metric breakdowns and additional analysis will be included in the revision.
>
> | **Agents** | **EduMirror (Avg. of 4 metrics)** | LLMob (Avg.) | BabyAGI (Avg.) | D2A (Avg.) | ReAct (Avg.) |
> | --- | --- | --- | --- | --- | --- |
> | 5 | **4.80** | 4.25 | 4.10 | 3.35 | 2.35 |
> | 15 | **4.18** | 3.60 | 3.57 | 3.53 | 2.93 |
> | 30 | **4.03** | 3.83 | 3.86 | 3.12 | 2.41 |
>
> The result shows that EduMirror supports continuous educational dynamics in realistic settings. Scaling the number of agents leads to roughly linear growth in simulation time and API cost, while context length remains bounded via our memory design (windowing and retrieval), avoiding context explosion. Since interactions are sparse and localized (e.g., small-group), only a subset of agents is active at each step, enabling efficient modeling of key interaction structures without requiring fully connected large-scale simulation.
>
> > ***Q2 (Validation of causal conclusions):** How is causal validity ensured and aligned with empirical research?*
> >
>
> EduMirror validates causal relationships through parallel simulation branches under controlled initial conditions. When evaluating an intervention (e.g., a teacher strategy), all other factors remain fixed, isolating it as the sole independent variable. The causal impact is then quantified via our Dual-Track Measurement Protocol, which compares behavioral metrics and internal states across branches.
>
> To validate the outcomes, we compare simulation results with real-world empirical studies. In Case Study 1, the "supportive-cooperative" intervention is most effective for victim recovery, consistent with findings that cooperative support outperforms punitive or neglectful approaches [6,7]. In Case Study 2, unregulated elections amplify rivalry, whereas fairness-oriented interventions promote balanced cooperation, aligning with sociological evidence on competition and inequality [8,9].
>
> **References:**
>
> [1] Liu, Y. et al. G-Eval: NLG Evaluation using GPT-4. EMNLP 2023.
>
> [2] Shi et al. Position Bias in LLM Judges. IJCNLP 2025.
>
> [3] Zheng et al. MT-Bench & Chatbot Arena. NeurIPS 2023.
>
> [4] Piaget, J. The Origins of Intelligence in Children. 1952.
>
> [5] Kohlberg, L. Stage and sequence: The cognitive-developmental approach to socialization. 1969.
>
> [6] Bilz, L. et al. Gewalt und Mobbing an Schulen. 2017.
>
> [7] Wachs, S. et al. Bullying intervention in schools. J. Early Adolesc., 2019.
>
> [8] Krupp, D. B. & Cook, T. R. Local competition and inequality. Psychol. Sci., 2018.
>
> [9] Killen, M. et al. Equity and justice in developmental science. Child Dev., 2016.

---

> > ### Author Rebuttal · Reviewer_Nqz4 · 2026-04-05
> >
> > I thanksthe authors for their detailed reply.

---

### Decision · Program_Chairs · 2026-04-30

**Decision:**

Accept (regular)

**Comment:**

Reviewers agree the multi-agent simulation platform for educational social dynamics, like bullying, cooperation, classroom interactions, is well-designed, based on established social and psychological theories and including interventions, and has potential for scaling up studies that are difficult to conduct in the real world. There are some concerns surrounding LLM-as-judge evaluations and simulation complexity. These were at least partially addressed during the response period, so authors should be sure to include these additional evaluations and discussions in the revision.